

# Where curling stones collide with rock physics: Cyclical damage accumulation and fatigue in granitoids

Derek D.V. Leung[1,2,3*], Florian Fusseis[1,4*], and Ian B. Butler[1]

[1]School of GeoSciences, University of Edinburgh, Edinburgh, UK
[2]Harquail School of Earth Sciences, Laurentian University, Sudbury, Canada
[3]Department of Earth Sciences, University of Regina, Regina, Saskatchewan, Canada
[4]Applied Structural Geology, Rheinisch-Westfälische Technische Hochschule RWTH, Aachen, Germany

**Correspondence:** Derek D.V. Leung (derekdvleung@gmail.com) and Florian Fusseis (florian.fusseis@rwth-aachen.de)

**Abstract.** Fatigue and damage accumulation in granitoids are classical, but poorly characterised, rock mechanics problems. We explore these phenomena by examining curling stone impacts. Curling stones are slid on ice and made to collide along a circumferential striking band. This well constrained scenario involves uniaxial compression of convex surfaces (i.e., Hertzian contacts). Each stone experiences about 2900 impacts per season, over a lifespan of 10–15 years before refurbishment, provid-

ing a unique opportunity to study fatigue and damage accumulation under cyclic loading.

Here, we first determine the stress magnitudes of head-on curling stone impacts using on-ice experiments involving a high-speed camera and pressure-sensitive films. We then characterise the damage observed in aged stones using photogrammetry, microtomography, and microscopy. For high-velocity impacts ($2.93\pm0.15$ ms$^{-1}$), a curling stone is locally and momentarily stressed to at least 300–680 MPa, exceeding its unconfined compressive strength. Curling stone impacts are dynamic in nature,

as evidenced by strain rates ($24\pm4$ s$^{-1}$) that resemble seismic magnitudes, ejection of rock powder during collisions, and prevalence of Hertzian cone fractures in aged stones. In the striking band, damage is confined to macroscopic Hertzian cone fractures and their immediate collet zones, and does not appear to extend beyond about 3–5 cm into the stones (radially). The circumferential density of cone fractures is limited to about 2–2.5 fractures per cm.

We propose that (1) early, high-velocity impacts initiate cone fractures up to a specific spatial density, and (2) with subsequent

collisions in the same regions of the striking band, cone fractures progressively propagate and coarsen. This concentrates and channels the accumulated damage, shielding the rest of the stones from reaching critical stress levels for damage. Our findings are significant for applications where rocks are exposed to repetitive, high-stress impacts and suggest that narrow damage zones can dampen high-impact stresses.

## 1 Introduction

Mechanical impacts cause permanent damage when their associated stresses exceed a material's fracture strength. Where these impacts occur repeatedly, they lead to fatigue and an irreversible deterioration of a material's mechanical properties. Understanding the damage and fatigue of a material is an important condition for its use in engineering applications. Where these applications involve natural rocks, the characterisation of damage and fatigue becomes a rock mechanics problem. In





fact, a great number of laboratory studies have addressed this problem for a wide range of natural rocks, often various types

of building stones (e.g., Cartwright-Taylor et al., 2020; Moore and Lockner, 1995; Moore et al., 1987). In rock mechanics, understanding how a rock responds to mechanical loads and how its strength evolves through repeated loading is important for predicting the effects of rock falls and earthquakes, but also for evaluating their performance in structures such as tunnels and bridges. Here we opt for an unconventional approach to study damage and fatigue in rocks: we use the collision of curling stones to study damage evolution and fatigue in granitoids.

Curling is an Olympic winter sport in which athletes slide stones across a sheet of ice, aiming to eject the competitors' stones from a target area (Fig. 1a). Viewed from a geologist's perspective, the case of colliding curling stones poses a perfectly defined long-term rock mechanics experiment. Sample materials and dimensions are well known, with most curling stones coming from two locations in the UK and being machined to predefined standards (Leung and McDonald, 2022). During a game, the impact velocities and contact areas of colliding stones can be measured, and thus the impact stresses can be determined. Curling stones

are known to incur damage over their lifetimes, and eventually exhibit crescent-shaped fractures along their circumferential striking bands (Fig. 1b). After having been played for 10–15 years and having experienced about 2900 collisions/year, every centimetre of a curling stone's striking band has experienced over 300 impacts. When curling stones are eventually retired, they allow for a detailed characterisation of the resulting effects on the rock macro- and microstructure. An experimental study of curling stone collisions offers an opportunity to quantify the impact process, and to explore how damage emerges and manifests

itself in curling stones.

Here, we combine on-ice collision experiments and detailed post-mortem analyses on retired stones to describe how curling stones from Ailsa Craig (Firth of Clyde, Scotland) are surprisingly effective in withstanding multiple high impact events. Impact stresses up to 680 MPa and delivered with seismic strain rates are dissipated over macroscopic damage structures penetrating no deeper than 3–5 cm into the stones, even after many thousands of collisions. The resulting model for damage

evolution in curling stones may be extrapolated to comparable scenarios in a geological or engineering context.

## 2   Mechanical background for characterising curling stone collisions

The stress of a curling stone impact can be calculated in two ways: (1) using an impact mechanics approach and (2) using a contact mechanics approach (Leung, 2020; Ling et al., 2018; Barber, 2018).

Following the impact mechanics approach, the average stress of the collision $\sigma_{avg}$ can be calculated based on the average

force $F_{avg}$ and collisional area $A$ (Ling et al., 2018):

$$\sigma_{avg} = \frac{F_{avg}}{A}. \tag{1}$$

The average force, in turn, can be determined by Newton's second law:

$$F_{avg} = ma = \frac{dJ}{dt} = \frac{mv' - mv}{\Delta t} \tag{2}$$

where $m$ = mass, $a$ = acceleration, $J$ = impulse, $\Delta t$ = contact time, and $v'$, $v$ represent the final and initial velocities (respec-

tively). The mass, velocity, and contact time can be measured from the on-ice experiments.





**Figure 1.** (a) Curling stones collide with one another during a game (©Tom Rowland / World Curling Federation). (b) These collisions cause the accumulation of damage in the circumferential striking bands of curling stones, which manifests as crescent-shaped fractures, flattened to concave profiles, pitting, and compound chipping.





As an extension of the mechanical parameters, the average elastic strain $\varepsilon_{avg}$ and strain rate $\dot{\varepsilon}_{avg}$ can also be approximated, provided that the Young's modulus $E$ of the material is known (Ling et al., 2018):

$$\varepsilon_{avg} = \frac{\sigma_{avg}}{E} = \frac{\Delta L}{L_0}, \tag{3}$$

where $\Delta L$ is the change in the rock's length during deformation and $L_0$ the undeformed length; and

$$\dot{\varepsilon}_{avg} = \frac{\varepsilon_{avg}}{\Delta t}. \tag{4}$$

The contact mechanics approach assumes the static or quasi-static loading of two curved surfaces, modelled as prolate ellipsoids with combined curvatures $\kappa_a$ and $\kappa_b$ (i.e., Hertzian contacts). The contact area of these two ellipsoids generates a contact ellipse with semi-major and semi-minor axes $a$ and $b$ (where $a > b$). The mean stress $\sigma_{avg}$ can be expressed with respect to the maximum stress $\sigma_{max}$ (Barber, 2018):

$$\sigma_{avg} = \frac{2}{3}\sigma_{max}. \tag{5}$$

The maximum stress $\sigma_{max}$ can be determined if the dimensions of the contact ellipse ($a$ and $b$) and one of the two curvatures ($\kappa_a$ or $\kappa_b$) are known (Barber, 2018):

$$\sigma_{max} = \frac{\kappa_a E^* a^2 e^2}{[K(e) - E(e)]b} \tag{6}$$

$$\sigma_{max} = \frac{\kappa_b E^* a^2 e^2}{\left[\frac{E(e)}{1-e^2} - K(e)\right]b}. \tag{7}$$

where $e$ is the eccentricity, and $K(e)$, $E(e)$ represent the complete elliptic integrals of the first and second kind, respectively, and $E^*$ is the composite Young's modulus which takes into account the Poisson ratio ($\nu$) of the material:

$$\frac{1}{E^*} = \frac{1-\nu_1^2}{E_1} + \frac{1-\nu_2^2}{E_2}. \tag{8}$$

## 3  Materials and methods

We studied curling stones from both Ailsa Craig (Firth of Clyde, Scotland) and Trefor (Llŷn Peninsula, North Wales), but here we focus on the former. Two types of rocks from Ailsa Craig are used in curling stones: Ailsa Craig Common Green, which is currently used for the striking bands of Olympic-standard curling stones; and Ailsa Craig Blue Hone, which was used as the striking band in older stones, but is currently inserted into the running bands of the stones (Leung and McDonald, 2022; see Fig. 1b for locations of the running band and striking band).

### 3.1  On-ice experiments

A series of 30 on-ice experiments was devised at Curl Edinburgh (Edinburgh, UK) in order to determine the mechanical parameters of curling stone impacts. The nine experiments we report here were independent head-on collisions between a



stationary stone and a moving stone that were delivered full-length by the 2025 World Men's curling champion and 2022 Olympic silver medallist Bruce Mouat. These Olympic-standard stones were composed of Ailsa Craig Common Green striking

bands inserted with Ailsa Craig Blue Hone running bands, and were in new condition, having been used for less than one season. While the same two stones were used in all experiments, the velocity of the moving stone was systematically varied between the experiments. In all experiments, a stopwatch was used to determine the velocity frame of reference for curling, and regular videography was employed for a velocity analysis. Some experiments used high-speed videography to determine the contact time, whereas others used pressure-sensitive films and aluminium foil to measure the contact area (Fig. 2).

In curling, the velocities of curling stones are compared using stopwatches by recording the travel times of curling stones over distances marked by lines. The hog-to-hog time is a common frame of reference; this time interval is measured between the two hog lines that separate the two playing ends by a distance of 21.945 m World Curling Federation (2018). The typical hog-to-hog times for takeouts range between 6.5–12 s; thus, this range of velocities was used in the experiments. The 6.5 s dataset is referred to as the maximum velocity scenario, although greater velocities are achievable in practice.

Two GoPro Hero 4 cameras recorded the kinematic (i.e., position and velocity) data of curling stones at 240 fps, using WVGA resolution (800 × 400 pixels). These cameras were positioned on an aluminium frame above the stationary stone ∼ 0.5 m from the ice surface. The data were processed as a series of image frames, and were corrected for fish-eye lens distortion using the Camera Calibrator App in Matlab® with a series of oriented calibration checkerboard images. The undistorted frames were then transformed to orthographic frames (with real-world pixel dimensions) using a homography transform, with

a calibration checkerboard as reference. The orthographic frames were analysed in Fiji (Schindelin et al., 2012; see File S1 in the Supplement for the image processing workflow). The positions of curling stones were marked by visually fitting circles to the handles of the stones and recording the coordinates of the centres of the circles. The precision of the kinematic data was determined by comparing reprojections of the calibration checkerboard to actual checkerboard dimensions, along with comparing the measured dimensions of the handles and curling stones (see File S2 in the Supplement for the error determination

of positional kinematic data). No significant systematic error was produced by the correction of fish-eye lens distortion. To account for potential errors in the kinematic data, the $2\sigma$ values derived from this analysis were used for error propagation.

A high-speed camera (Photron Fastcam SA 1.1 Colour) was used to record the collisions and their respective contact times. The camera was equipped with a macro lens with no barrel distortion and recorded collisions at selected frame rates ranging between 10–40 kfps; the final dataset presented here was recorded at a frame rate of 40 kfps. The high-speed camera was

triggered by external input, using a microphone sensor with an Arduino Uno interface. The data were processed as a series of still image frames, and a Sobel filter was applied in Fiji (Schindelin et al., 2012). The resulting images were used to determine when the stones were in collision.

Aluminium foil and pressure-sensitive film (Fujifilm HHS) were taped onto the striking bands of the stationary curling stone to determine the contact area and pressure distribution of the collisions. These films were subsequently scanned using

a document scanner at 600 dpi, and the contact area and dimensions (i.e., width and height) were measured visually via Fiji (Schindelin et al., 2012).







**Figure 2.** High-speed camera experiment set-up.





Preliminary observation of the high-speed camera data showed evidence for the ejection of rock powder during curling stone collisions. We used a toothbrush to gently remove powdered material from the striking band of a currently used curling stone for analysis with a scanning electron microscope at the School of Geosciences at the University of Edinburgh.

## 3.2    Macroscopic damage characterization

The morphology of crescent-shaped fractures was documented using a combination of photography and 3D scanning of sections of aged striking bands from Leung (2019) and an additional, full Ailsa Craig Blue Hone stone (Fig. 1a). The shape and size of crescent-shaped fractures were determined by digitising the outlines of crescent-shaped fractures from photographs using equally spaced points in Fiji (Schindelin et al., 2012). To determine the radius of curvature of the fractures, the points

were fit to a circle using least-squares regression by minimising the error between observed and calculated radial distances. Other curved fits, including ellipses and superellipses, were tested; however, these fits were found to be unsuitable because the fractures are not closed curves. For partially developed crescent-shaped fractures, the y-axis centre was fixed to the centre of the striking band in order to prevent mathematically non-sensible results. The least-squares fit thus models the fractures as circular arcs, which are each defined by a radius of curvature and central angle (here called the angular length).

The distribution of crescent-shaped fractures was determined from a full, aged Ailsa Craig Blue Hone curling stone. This curling stone was placed on a rotating table and was photographed in 34 sections, with each section representing $\sim 11°$. These sections were then manually aligned to form a rolled out cylinder of the striking band. This resultant panoramic image was analysed using digital image analysis in the Fiji environment (Schindelin et al., 2012). To determine the density and distribution of crescent-shaped fractures, a horizontal transect line was drawn across the middle of the striking band, and the intersection of

the crescent-shaped fractures with the horizontal transect line was recorded. Some incipient fractures did not directly intersect the transect line, although these commonly had visible components above and below the transect line. In these cases, the incipient crescent-shaped fractures were projected to intersect the transect line. The crescent-shaped fractures were classified by orientation (left- or right-convex) in order to evaluate any preferences in orientation.

The 3D morphology of crescent-shaped fractures was documented by (1) structured-light 3D scanning, (2) structure-from-

motion photogrammetry, and (3) synchrotron-based X-ray microtomography (SμCT). Structured-light 3D scanning was conducted with an EINSCAN Pro 2X Plus in fixed scan mode ($\sim 40$ μm resolution). For structure-from-motion photogrammetry, photos of varying orientation were taken with a Samsung Galaxy Edge 7 and processed using Meshroom (AliceVision, 2018) to produce 3D models of the samples. To look at the distribution of large-scale damage structures and microfractures within the samples in 3D, one sample was scanned using SμCT at the Advanced Photon Source, beamline 2-BM (Micro-tomography)

with a 30 keV pink beam. The scan consisted of 3600 projections (0.05°steps) with a 0.05 s exposure time, with a projection size of $3456 \times 1202$ pixels, and a pixel size of 3.13 μm. The projections were reconstructed using TomocuPy (Nikitin, 2023). Segmentation of fractures was conducted using the multi-scale Hessian fracture filter (Voorn et al., 2013).





### 3.3 Microscopic damage characterisation

In order to determine the microscale damage of curling stones, image mosaic transects measuring $\sim 1.5 \times 4$ mm were col-
lected from polished thin sections made from radial cuts through pristine and aged Ailsa Crag Blue Hone. Backscattered elec-
tron image mosaics (SEM-BSE) were acquired using a Carl Zeiss SIGMA HD VP Field Emission SEM under the following
conditions: $2000 \times$ magnification, 15 kV accelerating voltage, 30 $\mu$m aperature size, and 6.9–7.1 mm working distance. The
image mosaic transects were divided into $4 \times 4$ tiles and segmented for voids, quartz, albite, alkali feldspar, arfvedsonite, and
high-field-strength-element (HFSE) minerals in Fiji (Schindelin et al., 2012) using the Trainable Weka Segmentation plugin
(Arganda-Carreras et al., 2017). This segmentation yielded probability maps, which were segmented by binary thresholding at
a probability of $\geq 0.5$. Higher probability thresholds were attempted, but produced $> 10$ % unassigned pixels in some tiles (see
File S3 in the Supplement for probability threshold analyses of selected tiles). Initially, alkali feldspar and end-member albite
were segmented separately; however, due to the variable composition of alkali feldspar, this resulted in a significant number of
unassigned pixels. Thus, alkali feldspar and albite were not distinguished, and the sum of alkali feldspar and albite probability
maps was used for image segmentation analysis.

The phase map for voids represents a combination of fractures and pores, so an attempt was made to distinguish fractures
from pores. Generally, fractures have a larger perimeter-to-area ratio. Thus, the phase maps for voids were further subdivided
based on circularity ($c$):

$$c = \frac{4\pi A}{P^2} \tag{9}$$

where $A$ = area and $P$ = perimeter of void. A threshold of $c = 0.1$ was chosen by visual comparison; however, this subdivision
could not distinguish between fractures and complex porosity (often associated with feldspar phenocrysts), and thus the $c =$
0.0–0.1 subdivision reflects a combination of fractures and complex pores. The void ($c = 0.0$–$0.1$) phase map was used to
produce a damage profile for damaged and pristine samples in Fiji (Schindelin et al., 2012).

## 4 Results

### 4.1 On-ice experiments

Of the 30 experiments we conducted in total, we report 9 experiments here (kinematic and contact-area data are reported in
File S4 in the Supplement). The impact velocity was varied between 0.5—2.9 ms⁻¹ to determine how contact area and stress
vary with impact velocity (see the methods section for details of the on-ice experimental configuration).

Figure 3 shows a sequence of high speed camera frames depicting one of the collisions, with the stationary stone visible at
the bottom edge of the frames. For a radial impact velocity of $2.7 \pm 0.2$ ms⁻¹ the contact time was found to be $\leqslant 23 \pm 1$ frames
(frames 331–354) at 40 kfps or $\leqslant 0.575 \pm 0.025$ ms. Frames 354 and 400, which capture the separation of the two stones,
show that the collision also leads to the production and ejection of rock powder. An analysis of this rock powder using SEM-
BSE (Fig. 4) and energy-dispersive X-ray spectroscopy showed that most of the particles are monomineralic, and revealed a




**Figure 3.** Selected frames from high-speed camera experiment recording maximum-velocity scenario at 40 kfps (Exp. 2.13). Key frames are shown at different intervals: (a) start of experiment, (b) pre-collision (300), (c) earliest estimate of collision start, (d–e) mid-point of collision, (f) latest estimate of collision end, (g) post-collision and (h) end of data collection (see Video S1 in the Supplement for the unabridged video).




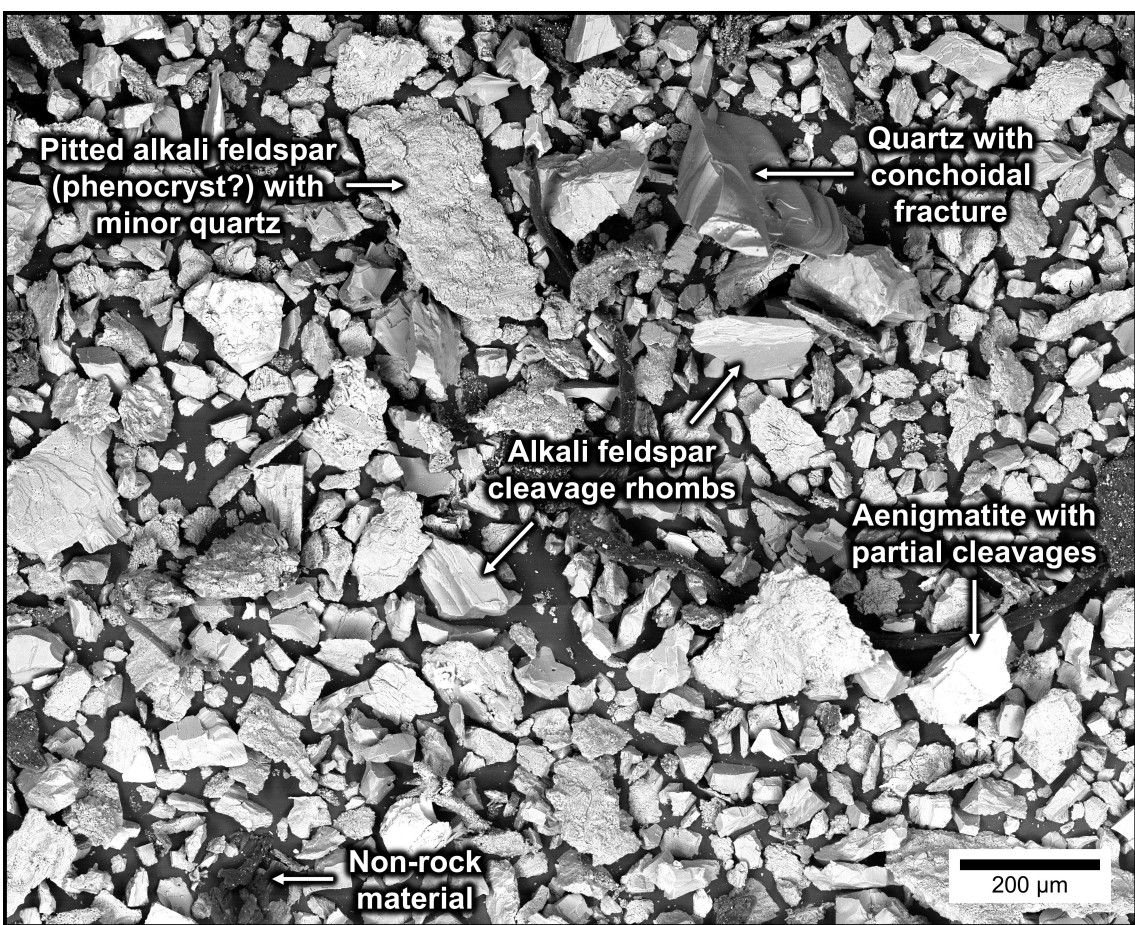

**Figure 4.** Representative backscattered electron (SEM-BSE) image of the rock powder from the striking bands of the curling stones.

distribution of fragment sizes with larger components formed by conchoidal drusy quartz and alkali feldspar phenocrysts, with

finer grains formed by minerals with a good cleavages (e.g., feldspars, pyroxenes, and amphiboles).

The contact areas have superelliptical shapes that result from the geometry of the striking band and its vertical extent. They scale with the impact velocity, where higher velocities cause larger contact areas (Fig. 5a-g). The larger contact areas reflect greater deformation and thus are indicative of higher contact stresses. We used pressure-sensitive film (Fujifilm HHS) to measure the contact stress for one experiment (Fig. 5h). In this film, the degree of colour saturation indicates the pressure in

static loading scenarios. The film we used was calibrated to 300 MPa and is fully saturated in our experiments, yielding a lower bound for the contact stress of 300 MPa. As a note, some caution must be advised on this estimation method, given that the pressure-sensitive films are calibrated for static loading conditions (as opposed to the dynamic loading conditions of curling stone impacts).



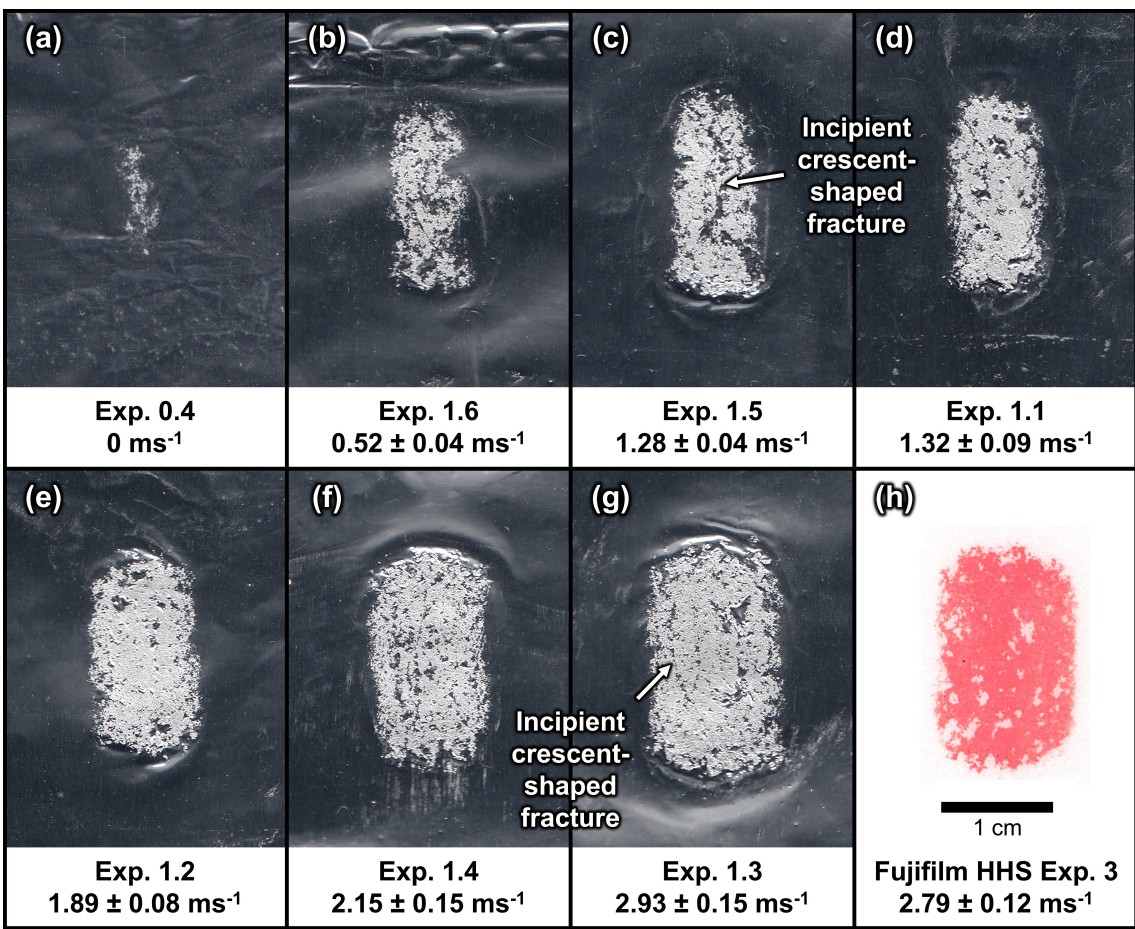

**Figure 5.** Contact areas recorded from various on-ice experiments. (a–g) Contact areas recorded on aluminium foil at increasing normal velocities: (a) 0 ms$^{-1}$, (b) 0.52±0.04 ms$^{-1}$, (c) 1.28±0.04 ms$^{-1}$, (d) 1.32±0.09 ms$^{-1}$, (e) 1.89±0.08 ms$^{-1}$, (f) 2.15±0.15 ms$^{-1}$, (g) 2.93±0.15 ms$^{-1}$. (h) Contact recorded on Fujifilm HHS pressure-sensitive film (at 2.79±0.12 ms$^{-1}$) shows fully saturated contacts, indicating stresses in excess of 300 MPa.





Using equations 5 to 7 we can calculate the impact stresses. For the maximum impact velocity of $2.93 \pm 0.15$ ms$^{-1}$, we determined an average stress over the contact area $\sigma_{avg}$ of 680 MPa (with a minimum error on the order of $\sim 30$ MPa based on the estimate of Poisson's ratio; Leung, 2020). Alternatively, using the impact mechanics approach (equations 1 and 2), we arrive at an average stress over the duration of the collision of $550 \pm 80$ MPa, which agrees with the former approach (see File S5 in the Supplement for calculations). For the same impact velocity we also calculated the minimum strain and strain rate using equations 3 and 4, assuming for the sake of simplicity that all strain is accommodated elastically, and a Young's modulus value of 39 GPa (Leung, 2020). This calculation yields a minimum strain of $1.4 \pm 0.2$ % and a minimum strain rate of $24 \pm 4$ s$^{-1}$.

## 4.2 Macroscale damage characterisation

### 4.2.1 Morphology of crescent-shaped fractures

Aging curling stones made from Ailsa Craig granite are known to develop crescent-shaped fractures along their striking bands. These fractures also occur in stones from other localities (e.g., Trefor quarry, North Wales) but are less commonly observed. The stone shown in Fig. 1b has likely been played for several decades and exhibits a severely damaged circumferential striking band, typifying the macroscopic damage that curling stones incur over their lifetimes. Crescent-shaped fractures represent the dominant damage type along with flattening of the striking band. At first observation, crescent-shaped fractures are 2–3 cm long, curvilinear fractures that extend to some depth into the curling stone. Flattening of the striking band is visible over time, as the striking band profile is typically convex for new stones, but becomes progressively concave as the stones age. Concave striking bands commonly develop undesired chipping and widening of the striking band margins. The preferential pitting of phenocrysts, as well as compound chipping from intersecting crescent-shaped fractures represent lesser damage features. Here, we focus mainly on characterizing the crescent-shaped fractures, which stand out due to their distinctive morphology.

Our macroscopic characterisation, which uses photogrammetry, structured-light 3D scanning (Fig. 6a–c) and SμCT imaging (Fig. 7d) combined with digital image analysis, reveals details on these fractures. Crescent-shaped fractures are convex in both trace and depth profiles, with a morphology approximating a paraboloid or conoid. The fracture surface shown in Fig. 6a–c has a pearly lustre and has corrugations which emanate from the striking band. This sample shows that crescent-shaped fractures can extend at least as deep as 1.5 cm from the striking band (in general penetrating to a maximum of 3–5 cm). A SμCT scan of a similar, but unexposed fracture (Fig. 7c) displays similar features, and, most interestingly, confirms that the damage is restricted to the crescent shaped-fracture and associated subsidiary damage in their collet zone, which is the wedge between the crescent shaped fracture and the striking band (Fig. 7c).

### 4.2.2 Macroscale evolution of the shape and distribution of crescent-shaped fractures

Crescent-shaped fractures are discrete, spaced fractures, which have have two distinct orientations (left- or right-convex; Fig. 1b). None of the fractures appear to be closed forms (*i.e.*, complete ellipses or circles). Crescent-shaped fractures range in their degree of development—both within individual samples and between different samples—from incipient (partially formed



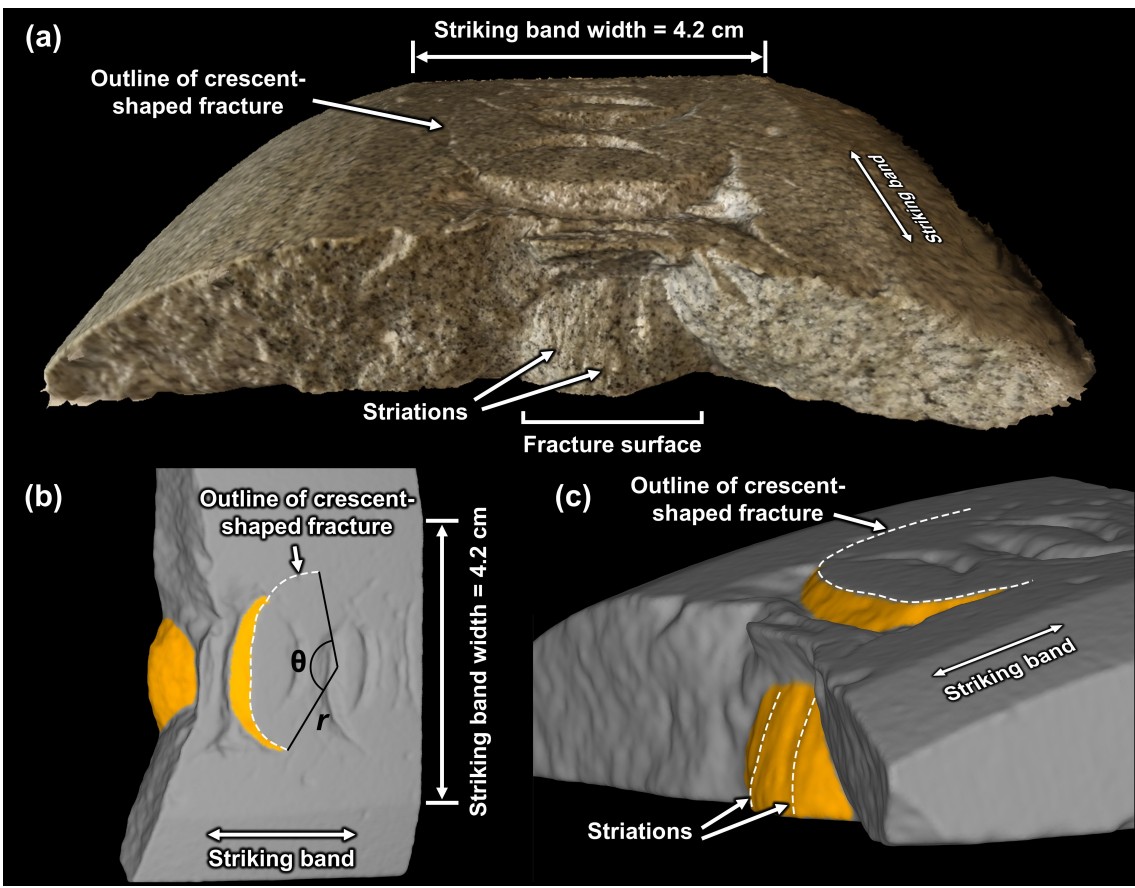

**Figure 6.** Photogrammetric (a) and structured-light (b-c) 3D scans of an Ailsa Craig Blue Hone striking band (AC-01), in which the 3D morphology of an isolated, mature-stage crescent-shaped fracture is exposed (note that the majority of fractures in this sample are juvenile). (a) Oblique radial view of striking band (with color and topography); the surface of the crescent-shaped fracture has a pearly lustre and is striated. (b) Tangential view of striking band (topography only), showing the orientation of the crescent-shaped fracture (orange) with respect to the striking band; the statistical analyses for radius of curvature ($r$) and angular length ($\theta$) are illustrated. (c) Oblique profile view (topography only) showing that the crescent-shaped fracture resembles a parabaloid, i.e., it is convex in three dimensions. The striations on the surface of the fracture emanate from the striking band.







**Figure 7.** Synchrotron-based microtomographic scan of an Ailsa Craig Blue Hone sample (thin-section cutoff) with juvenile to mature crescent-shaped fractures (AC-03-2). (a) 3D scan overview of sample, with locations for subfigures b, c, and d. (b) Top view of sample, showing the viewing orientation for c. (c) Results from multi-scale Hessian fracture filtering (Voorn et al., 2013), showing fractures and void spaces; note in particular the orientation of the crescent-shaped fracture, damage in the collet zone, as well as porosity detected in alkali feldspar phenocrysts and in the groundmass. (d) Representative tomograph showing the differences in attenuation between various minerals with respect to fractures and porosity.



**Figure 8.** Digitised crescent-shaped fractures of selected sections of striking bands, normalized to the centre of curvature, and presented in order of increasing damage state: (a) Red Trefor sample TF-02, (b) Ailsa Craig Common Green sample AC-10, (c) Ailsa Craig Blue Hone sample AC-01, and (d) Ailsa Craig Blue Hone sample AC-14. In each subfigure, the whole data set is shown in grey to contextualize relative distributions in morphology. (e) Statistics of the digitised crescent-shaped fractures, with the radius of curvature ($r$) showing less variation with damage state, and the angular length of the fractures ($\theta$) increasing with damage state.





**Figure 9.** Distribution of macroscopic crescent-shaped fractures from various aged striking band samples. Left-convex fractures are marked with blue opening parentheses, whereas right-convex fractures are marked with red closing parentheses. Sample TF-02 (Red Trefor, very weakly incipient) is not shown due to the lack of intersection data. Symbols and abbreviations: lc = left-convex, rc = right-convex.




fractures, which may be formed in segments) and juvenile (fully linked crescent-shaped fractures, with 3D fracture surface hidden) to mature (fully linked crescent-shaped fractures, with 3D fracture surface exposed). Despite the typical crescent-shaped morphology of the fractures, irregular fractures also exist; these commonly represent incipient fractures or compound chipping due to the intersection of two fractures. Moreover, the fractures may consist of multiple, superimposed crescent-

shaped fractures that bifurcate towards the margins of the striking bands. These overprinting fractures are more common on mature striking bands (*e.g.*, Fig. 8d).

The radius of curvature of the crescent-shaped fractures varies between 0.5–2.5 cm, and does not appear to be markedly different between samples of different damage state or rock type (Fig. 8e). The interquartile range of the radii of curvature (illustrated by the width of the boxes on (Fig. 8e) decreases as the damage state increases; this may in part be related to the

variation in the sample size between each curling stone sample, with a smaller sample size correlating with greater variation. On the other hand, the angular length of crescent-shaped fractures (Fig. 8e) increases as a function of age, suggesting that crescent-shaped fractures grow over repeated impacts.

The total density of crescent-shaped fractures does not increase significantly after the damage state exceeds an incipient state (Fig. 9), although it should be noted that pristine to weakly incipient stones (such as TF-02) have distinctly lower fracture

densities. This suggests that the density of crescent-shaped fractures must increase between the weakly incipient and incipient damage states, at which point the striking band becomes saturated in crescent-shaped fractures.

The number of left- and right-convex crescent-shaped fractures is approximately equal in the observed samples (Fig. 9), suggesting that both have equal probability of forming. However, it is evident that there are domains dominated by left- or right-convex crescent-shaped fractures. In other words, left- and right-convex fractures do not form conjugate sets.

### 4.3 Microfracture descriptions

#### 4.3.1 Microscale anatomy of crescent-shaped fractures

In a radial view observed in thin section (Fig. 10a), crescent-shaped fractures approximate circular arcs. However, at the ends of the fractures, they become horizontally asymptotic in attitude (Fig. 10b). This horizontal asymptotic rollover is not observed for the equatorial sections of the crescent-shaped fractures imaged in 3D (Fig. 6), suggesting that crescent-shaped fractures are

not axially symmetric. Moreover, the crescent-shaped fracture shows minor undulations (Fig. 10a), which could represent a cross-sectional view of the striations reported in the macroscopic characterisation (Fig. 6). In the collet zone, which is located between the crescent-shaped fracture and the striking band, there is a narrow, localised damage zone, and towards the striking band, there are low-angle microcracks that may be produced by spalling (Fig. 10c). Behind the crescent-shaped fracture and towards the centre of the curling stone, the sample appears to be pristine. In the following subsection, we take a closer look at

the microcracks found in this sample using segmentation analysis of SEM-BSE images (Fig. 11).



**Figure 10.** (a) Radial section of striking band showing juvenile fractures under reflected light. (b) Near the margins of the crescent-shaped fracture, the curvature of the crescent-shaped fracture rolls over to become horizontally asymptotic. (c) Damage is strongly restricted to the collet zone (the collet zone is a wedge-shaped area located between the crescent-shaped fracture and the striking band). Some microcracks are visible near the surface of the striking band and may correspond to spalling.





### 4.3.2 Damage profile analysis

Segmented phase distribution maps derived from digital image analysis of SEM-BSE maps (Fig. 11a) illustrate the microscale characteristics of the crescent-shaped fracture and associated damage zone (Fig. 11b). A profile of voids with circularity between 0–0.1, which reflects fractures and complex zones of porosity, yields an approximate damage profile (Fig. 11c–d). This profile shows that the damage increases from the striking band to the main fracture and sharply diminishes to a pristine state towards the centre of the stone. Between the striking band and the main crescent-shaped fracture, the percentage of pixels relating to segmented fracture porosity exceeds that of the pristine sample (AC-12, blue line, Fig. 11d). On the other hand, towards the interior of the stone, the pixel density generally falls below that of the pristine sample (Fig. 11). Outliers within the interior of the stone are related to zones of complex porosity within feldspar phenocrysts (Fig. 11b–c). In total, the damage profile shows that the damage is strongly restricted to crescent-shaped fractures and localised damage zones.

Given that the damage is strongly restricted to crescent-shaped fractures and localised damage zones, the segmented map for AC-03-1 can be subdivided into damaged and pristine regions. Digital image analysis of the two regions shows a 5.3% increase in voids with circularity between 0–0.1 and a 4.5% increase in unclassified pixels for the damaged region, which correlates to an increase in microcracks. The ratio of alkali feldspar to quartz in the damaged region (6.4) is higher than that of the pristine region (4.9), suggesting that the abundance of quartz is lower in the damaged region. This could be due to (1) mineralogical heterogeneity across the analysed area or (2) preferential damage to quartz within the damage zone. On visual inspection of the damage zone, preferential damage to quartz is not observed, suggesting that mineralogical heterogeneity is responsible for the lower abundance of quartz in the damaged region.

### 4.3.3 Evolution of microfractures and their types

On the microscale, the progressive evolution of microfractures in increasingly damaged stones was visualized in Ailsa Craig Blue Hone samples from a pristine to a juvenile stage using reflected light microscopy and SEM-BSE imaging. These analyses show that even pristine samples of Ailsa Craig Blue Hone exhibit a level of pre-existing damage (Fig. 12a): intragranular and transgranular microcracks in quartz, grain-boundary microcracks between quartz and alkali feldspar/albite, cleavage and porosity-linked intragranular microcracks in alkali feldspar, and minor cleavage cracks in arfvedsonite (Blenkinsop, 2000). These microcracks are consistent with unloading and thermal contraction (thermally induced/elastic mismatch microcracking). Two types of porosity exist in alkali feldspar and affect later crack propagation: (a) large zones of complex porosity within phenocrysts, and (b) disseminated microporosity associated with patch perthite and secondary albite (Leung and McDonald, 2022).

In a sample with incipient damage, the crescent-shaped fracture is characterised by a series of coalescing transgranular microcracks with a maximum width of 0.6 $\mu$m (Fig. 12b). Most of the microcracks are flaw-induced and locally take advantage of quartz-alkali feldspar grain boundaries, as well as porosity and cleavages within alkali feldspar. The microcracks are intergranular where they link between microstructural features. Minor incipient damage exists in the collet zone as isolated, subparallel mode 1 fractures. The largest and most common microcracks in the damage zone are intragranular to marginally





**Figure 11.** Segmentation analysis of a SEM-BSE image for sample AC-03-1 (see location in Fig. 10). (a) original SEM-BSE image. (b) segmented map of various minerals and voids. (c) segmented maps of voids with circularity between 0–0.1. (d) Percentage of pixels classified as voids with circularity between 0–0.1 along the horizontal transect of b, which serves as a proxy for the damage profile. From left to right (going from the striking band towards the centre of the stone), the damage increases towards the crescent-shaped fracture and immediately reverts to pristine levels past the crescent-shaped fracture (see text for discussion).







**Figure 12.** Evolution of crescent-shaped fractures and microscale damage in Ailsa Craig Blue Hone curling stones (radial section). (a) Pristine samples contain randomly oriented, pre-existing microcracks. (b) Microcracks form parallel to the striking band and coalesce into an incipient crescent-shaped fracture. (c) The main crescent-shaped fracture widens and evolves into a microfault with visible gouge; a damage zone consisting of apparently subparallel, interconnected microcracks and microfaults develops in the collet zone between the striking band (direction indicated by arrow) and the main crescent-shaped fracture.





intergranular, porosity-linked, and/or cleavage microcracks within the alkali feldspar grains. Minor microstructural features

include intragranular microcracks within quartz grains and coincident grain-boundary microcracks between quartz and alkali feldspar, which locally extend to form intergranular microcracks.

In a more damaged juvenile sample (Fig. 12c), the main crescent-shaped fracture is expressed as a microfault with larger apertures between 3–45 $\mu$m. Importantly, this fracture contains what appear to be domains of fault gouge. This microfault is broadly coincident with quartz–alkali feldspar phase boundaries and grain boundaries between alkali feldspar grains, and can

be transgranular through alkali feldspar and quartz grains. However, the coarseness of the microfault generally obscures these microstructural relationships. The gouge within the main crescent-shaped fracture consists of (1) larger (up to 30 $\mu$m), locally derived, in-*situ* fragments, along with (2) smaller particles which have likely been rotated and displaced (Fig. 12c). The larger grains are almost entirely composed of alkali feldspar, whereas smaller grains are a combination of alkali feldspar, quartz, and albite, with rare arfvedsonite and high-field-strength-element minerals (note that among the smaller grains, quartz and

albite cannot be distinguished on the basis of their similar SEM-BSE responses). Where the crescent-shaped fracture cross-cuts arfvedsonite, the mineral can be found fragmented in the gouge zone, suggesting that the particles are locally derived. The collet zone exhibits a network of apparently subparallel, interconnected intra- and intergranular microcracks (Fig. 12c). These microcracks are heterogeneous and can be further subdivided into several types: (1) flaw-induced, porosity-linked in-tragranular microcracks in albite; (2) intragranular cleavage microcracks in alkali feldspar and arfvedsonite; (3) flaw-induced,

grain-boundary microcracks between alkali feldspar and quartz grains, or among feldspar grains; and (4) conchoidal, intra-granular impingement microcracks in quartz grains. Fracture networks are extensively developed in albite, possibly owing to the high degree of porosity within these grains.

## 5 Discussion

Our study explores a particular, potentially extreme configuration of cyclical loading in granitoids: tens of thousands of impacts

with dynamic loading, reaching stress magnitudes that exceed the compressive strengths of the material. This configuration differs fundamentally from more conventional loading and fatigue tests (Cerfontaine and Collin, 2018). A key difference is that the majority of the impact energy is invested in the acceleration of the resting stone on the ice. Nevertheless, the case of curling stone collisions is also peculiar in that the stones support these impacts for much longer than cyclical loading tests would suggest.

Determining the impact energy that is dissipated by the stone requires the exact quantification of the initial and final ve-locities of the moving and stationary stones. Although these were tracked during each experiment using the GoPro cameras, calculations of kinetic energy using these data yielded values of low precision. As a first-order approximation, the high-speed camera experiment no. 2.13 (stone mass 18.41 kg) yielded pre-impact velocities (and kinetic energies in parentheses) of 2.74 ms$^{-1}$ (69 J) and 0 ms$^{-1}$ (0 J) for the incoming and stationary stones, respectively. Immediately after the impact, these changed

to 0.09 ms$^{-1}$ ($<<$ 1 J) and 2.66 ms$^{-1}$ (65 J) for the incoming and now-accelerated stone. This implies an energy loss of $\sim$ 4 J ($\sim$ 6 %). This energy loss estimate also includes work done by friction while the stones slide on the ice surface, which





represents ∼ 1 J when applying a coefficient of friction of 0.1 (Nyberg et al., 2013), meaning that ∼ 3 J or ∼ 4 % of the impact energy is dissipated in the stone. The significance of these calculations is that, despite the fact that curling stones are subjected to high impact stresses, the stresses are instantaneously and mostly recovered by the acceleration of the resting stone. Our data suggest that the damage zone behind the striking band is efficient in dissipating the energy difference once it is saturated in fractures.

The aim of this contribution is to explore the effects of cyclical loading and fatigue on granitoids. In this section, we revisit the original goals of this contribution by (1) determining the boundary conditions of curling stone collisions and (2) understanding how damage accumulates in curling stones, with a focus on crescent-shaped fractures. We then conclude this section by (3) proposing a damage evolution model for curling stones that integrates the insights gained from the first two discussion points.

## 5.1 Boundary conditions of curling stone collisions

In terms of the boundary conditions of curling stone collisions, our results reveal that (1) curling stone impacts produce damage by incurring stresses that exceed the critical fatigue stress of the rocks; (2) that these stresses are significant and comparable to other geological phenomena; and (3) the impacts are dynamic in terms of their stress-strain regimes:

1. Our calculations from on-ice experiments indicate that the stress magnitudes of the impacts of curling stones (in excess of 300-680 MPa) exceed the local uniaxial compressive strength ranges recorded for curling stones (230–520 MPa; Nichol, 2001; Leung, 2020). Our post-mortem investigation indicates that these impact stresses are responsible for the observed damage. This is consistent with fatigue experiments (Zhou et al., 2018), which indicate that damage must exceed a threshold to produce fatigue damage.

2. The instantaneous stress magnitudes generated by curling stone impacts are comparable to those reached in geological processes and provide an opportunity to study damage evolution resulting from intermediate stress magnitude events. Table 1 shows that the stresses produced by curling stone impacts are larger than those induced by mining as well as those associated with co-seismic stress drops. They most closely resemble mid-crustal lithostatic stresses, rockfall, and thermally-induced stresses. Interestingly, natural, randomly oriented crescent-shaped fractures have been observed on boulders found on the coast of Ailsa Craig (fieldwork conducted by DDVL in Sep. 2020), possibly alluding to the similarity in deformation processes involved in rockfall versus curling stone impacts (Leung 2020). Crescent-shaped fractures are also common phenomena on glacier-bedrock interfaces, where high stresses emerge from the point loads exerted by boulders in the ice (Harris, 1943; Bestmann et al., 2006). On the other extreme of stress magnitudes, curling stone impacts produce much lower stresses than ballistic impacts and shock metamorphism.

3. The magnitude of strain rates in curling stone impacts ($24 \pm 4 \, \text{s}^{-1}$) suggests that there is a significant dynamic component involved in curling stone impacts, which is comparable to seismic events (Table 2). This is corroborated by the ejection of rock powder from the striking bands after impacts between curling stones, as observed by high-speed camera footage. Additionally, the presence of minor spalling microcracks in our post-mortem analysis also supports this interpretetation.



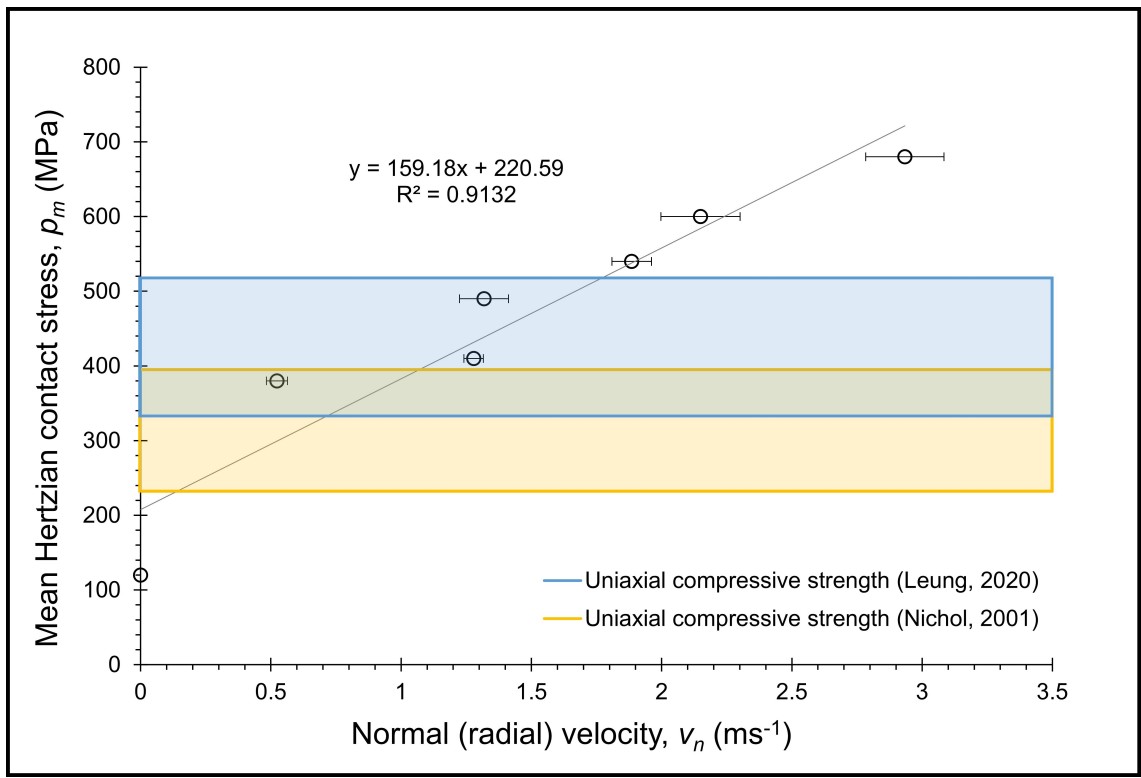

**Figure 13.** The stresses calculated from Hertzian contact analysis show that the high-velocity collisions can generate stresses which exceed the uniaxial compressive strengths of the rocks.

**Table 1.** Stress ranges for various types of deformation.

| Deformation type | Stress (MPa) | Reference |
|---|---|---|
| Mining-induced | $\sim 17$ | Ouyang et al. (2009) |
| Co-seismic stress drop | 2–200 | Onwuemeka et al. (2018) |
| Lithostatic (10–20 km) | $\sim 270$–540 | Fossen (2010) |
| Rockfall on concrete | $< 400$ | Mougin et al. (2005) |
| Thermal | $< 500$ | Schrank et al. (2012) |
| Ballistic impacts | $1.8 \times 10^4 - 1.18 \times 10^5$ | Ahrens and Rubin (1993) |
| Shock metamorphism | $< 3 \times 10^3 - > 1 \times 10^5$ | DeCarli (2005); Kieffer et al. (1976) |

Our microstructural observations might thus provide insights into the damage response of seismically loaded granitoids,
350    including pulverization.





**Table 2.** Strain-rate ranges for various types of deformation.

| Deformation type | Strain rate ($s^{-1}$) | Reference |
| --- | --- | --- |
| Geological background | $10^{-15} - 10^{-13}$ | Pfiffner and Ramsay (1982) |
| Rockfall on concrete | $10^{-1}$ | Mougin et al. (2005) |
| Co-seismic slip | $10^{1}$ | Fagereng and Biggs (2019) |
| Shock metamorphism | $10^{6} - 10^{9}$ | Stöffler and Langenhorst (1994) |

## 5.2 Crescent-shaped fractures form as Hertzian cone fractures

We interpret crescent-shaped fractures to be Hertzian cone fractures, on the basis of their correlation to the contact areas between colliding curling stones, as well as their conoid morphology. The contact areas measured from the on-ice experiments correlate with the size and shape of the crescent-shaped fractures (Fig. 14a), a feature exhibited by Hertzian cone fractures experimentally produced in other solid media (Lawn, 1998; Wang et al., 2017). Additionally, the conoid 3D geometry of crescent-shaped fractures is reminiscent of other features interpreted to be Hertzian cone fractures: glacial crescentic gouges (Fig. 14b), percussive fractures from fluvial environments (Fig. 14c–d), as well as experimentally generated fractures in glass (Fig. 14e). Based on this interpretation, we can use the predicted stress distributions around the Hertzian contacts to understand the observed distribution of microcracks in the striking band, and deduce how damage accumulated to form the crescent-shaped fractures.

The conoid 3D geometry of Hertzian cone fractures as displayed by crescent-shaped fractures is related to the $\sigma_1$ stress trajectories of Hertzian contacts (Kocer and Collins, 1998). The crescent-shaped fractures in our samples nucleate as intragranular cleavage cracks in feldspars that are oriented parallel to the predicted stress trajectories produced by Hertzian contacts, which subsequently link to form transgranular cracks. Similarly, in cyclical loading experiments, microcracks form with a preferential alignment parallel to $\sigma_1$, and the linking of intragranular cleavage cracks in feldspars represents the onset of unstable crack propagation (Chen et al., 2011; Akesson et al., 2004).

These initial damage structures are created by early, high-velocity collisions and controlled by the stress field trajectories of Hertzian contacts, with subsequent impacts propagating these initial damage structures. The early onset of these damage structures is evident from our observations during the on-ice experiments: the stones used in the experiment had been played for less than a season, yet they already displayed weakly incipient crescent-shaped fractures (Fig. 5). Furthermore, our post-mortem analyses show that the density of crescent-shaped fractures becomes saturated in the incipient stage of damage and does not increase towards the mature stage of damage. These observations are supported by acoustic emissions data recording cyclic loading experiments by Sondergeld and Estey (1981), which show that (1) the majority of acoustic emission events occur in the first cycle of loading, with fewer acoustic emission events in later cycles, and (2) acoustic emission hypocenters cluster around previous hypocenters. The implication is that significant damage structures are created during early collisions, with subsequent collisions propagating these damage structures.





We have also shown that the damage distribution is restricted to the crescent-shaped fractures and associated collet zones, with virtually no damage outside of these areas. This indicates that curling stones are extremely efficient at dampening damage by utilizing pre-existing structures, rather than forming new ones.

## 5.3 Damage evolution model for curling stones

We return to rock physics by proposing a multiscale damage evolution model of curling stones, which takes into account the cyclical damage conditions of curling stone impacts, as well as the macroscopic and microscopic damage features observed in curling stones (Fig. 15). The damage evolution of curling stones consists of several damage states: pristine, weakly incipient, incipient, juvenile, and mature.

*Pristine stage.* At the macroscale, no crescent-shaped fractures are visible. However, repeated curling stone impacts pulverise the surface of the striking band, resulting in the ejection of rock powder, progressive flattening of the striking band, and pitting. The flattening of the striking band occurs at the scale of ten to fifteen years with average use (largely within the pristine to incipient damage states), which coincides with the maintenance cycle of re-profiling curling stones. The duration of the pristine stage varies by rock type: some retired Blue Trefor curling stones have apparently remained in the pristine stage, whereas early stage evidence of crescent-shaped fractures has been recorded within less than a season of play in Ailsa Craig Common Green stones.

*Weakly incipient stage.* At the macroscale, the damage is difficult to discern but is characterised by sparse, partially developed, curvilinear discolouration features. Initially, high-velocity impacts initiate Hertzian cone fractures, whereas low-velocity impacts may propagate existing fractures. At the microscale, microcracks (mostly intragranular within feldspars) are distributed parallel to the predicted stress trajectories produced by Hertzian contacts.

*Incipient stage.* At the macroscale, crescent-shaped fractures are partially developed and are relatively visible, despite showing minimal topographic relief. The striking band is saturated with crescent-shaped fractures, so succeeding impacts propagate existing crescent-shaped fractures instead of creating new fractures. At the microscale, transgranular microcracks propagate to form through-going crescent-shaped fractures. A few microcracks mark a proto-damage zone between the striking band and the main crescent-shaped fracture.

*Juvenile stage.* At the macroscale, crescent-shaped fractures are well defined and develop topographic relief. Succeeding impacts cause crescent-shaped fractures to penetrate deeper into the curling stone, and a localised damage zone develops in the collet zone between the striking band and the crescent-shaped fractures. At the microscale, the crescent-shaped fractures widen to become through-going microfaults with gouge and continue to propagate at the margins of the fractures and into the stone. The damage zone consists of interconnected transgranular fractures.

*Mature stage.* At the macroscale, the fracture surface of crescent-shaped fractures is exposed on the striking band due to the ejection of material from the damage zone. As a result, the striking band becomes irregular and locally concave. Secondary structures such as compound chipping, exfoliation, and branched fractures develop. At the microscale, crescent-shaped fractures are expected to propagate deeper into the curling stone, while the damage zone grows laterally. Most curling stones are re-profiled or replaced before reaching this stage.



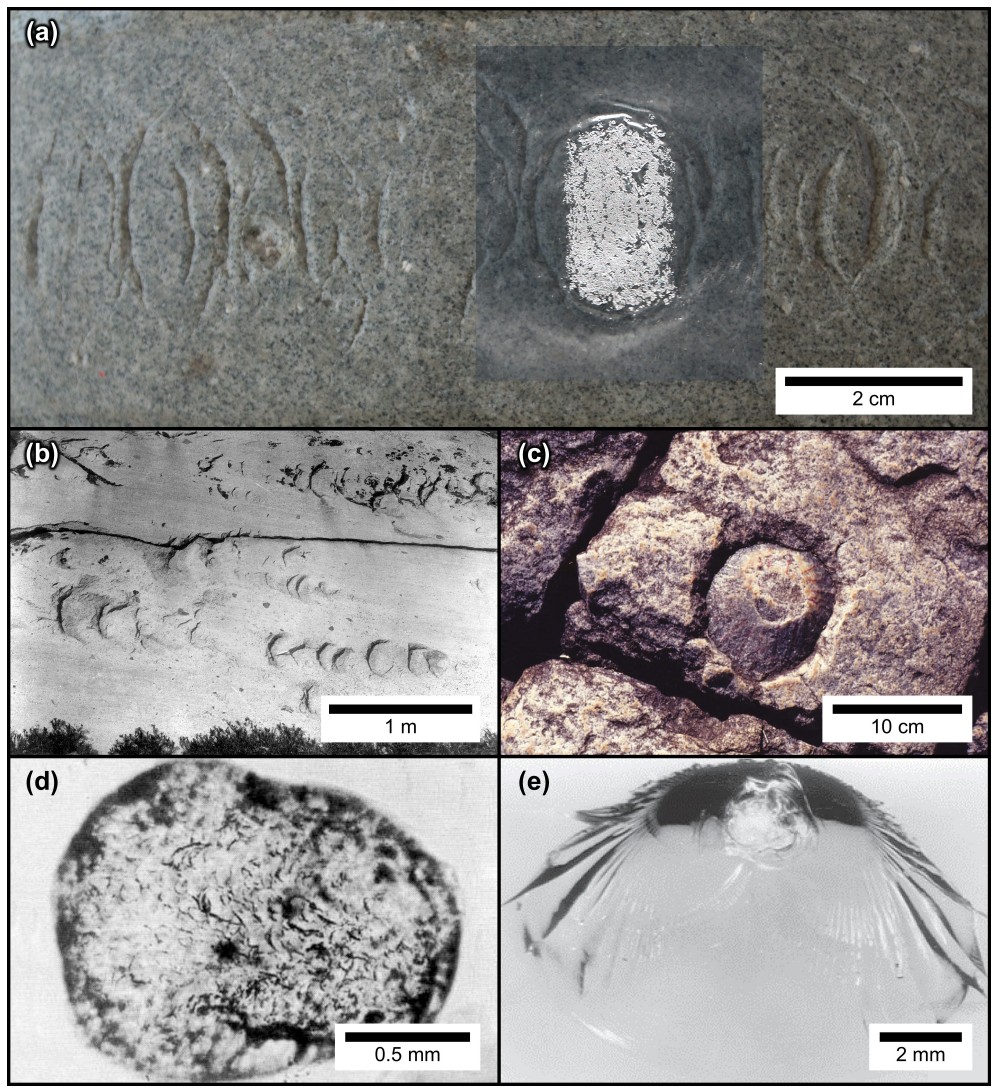

**Figure 14.** (a) Superimposing the contact area from the on-ice experiments (Exp. 1.3) onto an aged striking band (AC-01) shows the resemblance in shape and size between the contact areas and crescent-shaped fractures, suggesting that crescent-shaped fractures are Hertzian cone fractures. (b–e) Examples of natural and antropogenic Hertzian cone fractures. (b) Glacial crescentic gouges (a subclass of chatter marks) produced by point loading of erratics via glacial transport, Sierra Nevada, CA, USA (from Gilbert, 1906). (c) Percussive fractures produced by point loading of boulders in a fluvial environment (paleowaterfall), eastern circumference of Bushveld Complex, South Africa (from Reimold and Minnitt, 1996). (d) Quartz grain from sandstone showing abundant crescent percussion marks, northern kibble Co., TX, USA (from Campbell, 1963). (e) Experimentally generated Hertzian cone fractures in glass; note the presence of mirror-mist-hackle structures on the surface of the cone fracture (from Wang et al., 2017).



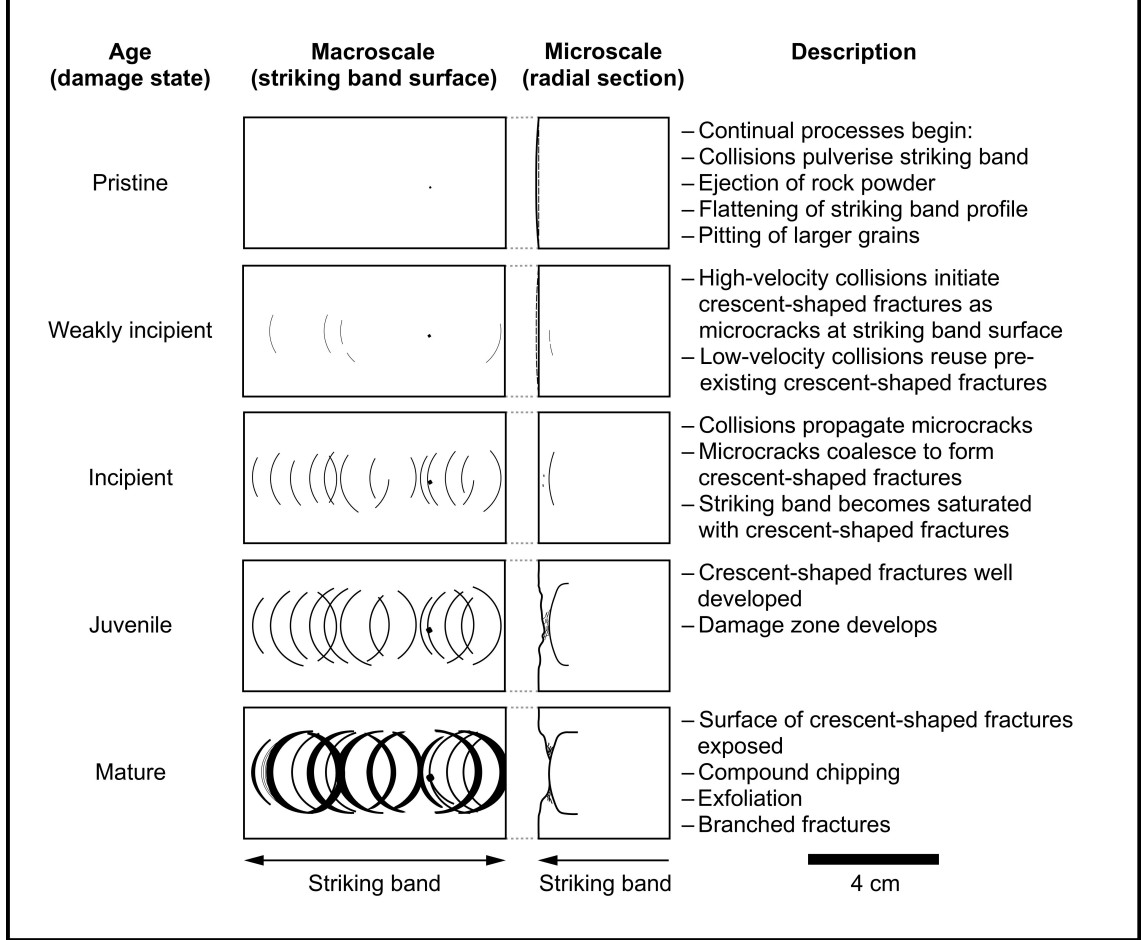

**Figure 15.** Damage evolution model for curling stones. Diagrams have no vertical exaggeration.

## 6 Conclusions

1. Curling stones develop damage from the stress of impacts exceeding the strength of curling stones. The magnitude of the strain rates resembles that of seismic events, indicating a dynamic component.

2. Crescent-shaped fractures are interpreted to be Hertzian cone fractures on the basis of their relationship to contacts between colliding curling stones, as well as their conoid morphology.

3. Damage structures develop early, mostly as intragranular microcracks in feldspars, which propagate into transgranular microcracks that eventually develop into through-going crescent-shaped fractures.




*Code and data availability.* Files S1 to S5 can be found in a GitHub repository linked to Zenodo: https://doi.org/10.5281/zenodo.16191540 (Leung et al., 2025). File S1 corresponds to GoPro image processing steps. File S2 corresponds to GoPro uncertainty analysis. File S3

corresponds to probability thresholds for image segmentation of sample AC-03-1. File S4 corresponds to kinematic and contact-area data for the on-ice experiments. File S5 contains the Matlab® script used to calculate Hertzian contact stresses.

*Video supplement.* Video S1 contains the high-speed camera footage and can be found on TIB AV-Portal: (doi to be assigned)

*Author contributions.* Contributions follow the CRediT model. DDVL: conceptualization, data curation, formal analysis, funding acquisition, investigation, methodology, project administration, resources, visualization, writing - original draft, writing - review and editing. FF:

conceptualization, funding acquisition, supervision, writing - review and editing. IBB: conceptualization, funding acquisition, methodology, resources, supervision, writing - review and editing.

*Competing interests.* Some authors are members of the editorial board of *Solid Earth*.

*Acknowledgements.* This contribution originates from an M.Sc.R. thesis completed by Derek D. V. Leung at the University of Edinburgh. Derek D. V. Leung was supported by the Scotland Saltire Scholarship, Mineralogical Association of Canada Foundation Scholarship, Young

Mining Professionals Yamana Gold Student in Mining Scholarship, and Gem and Mineral Club of Scarborough Jennifer and Blair Campbell Bursary. Research funding was supported by the Moray Endowment Fund, Edinburgh Geological Society, and the NERC CATFAIL project (NE/R001693/1). Fieldwork was supported by the Mykura Fund from the Edinburgh Geological Society. Synchrotron data were collected at Diamond Light Source (Oxfordshire, UK) under beamtime proposal MG22517 and Advanced Photon Source (Lemont, IL) under proposal 81226. Adam Griffiths, Terry Williams, and Bryn Griffith are thanked for providing access to the Trefor quarry. Mike Boyd and Anthony

Middleton (UCreate Studio, University of Edinburgh) are thanked for providing photogrammetry and electronics equipment. Andy Macpherson and Scott Henderson (Curl Edinburgh) are thanked for providing ice time for the on-ice experiments. Mark Johnson (Slowmo Ltd) and Alan Woolley (University of Edinburgh) provided access to high-speed cameras and lighting for the on-ice experiments. Bruce Mouat, Berit Schwichtenberg, and Dylan Price are thanked for their assistance with on-ice experiments. Nicola Cayzer assisted with SEM analyses. Godfrey Fitton and Alexis Cartwright-Taylor are thanked for their mentorship and support. Stephane Perrouty is thanked for access to a computer

workstation for processing the S$\mu$CT data.





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
