# Peer review of "Where curling stones collide with rock physics: Cyclical damage accumulation and fatigue in granitoids"

_EGUsphere, 2025_

## Author Comment (AC1)

**Reviewer 1**

*Review of Leung et al., 2025 in Geosphere: "Where curling stones collide with rock physics: Cyclical damage accumulation and fatigue in granitoids"*
*The authors aim to understand damage in granites subjected to cyclic loading. To do so, they see an opportunity in studying curling stones, made from granite and subjected to repeated impact loading. [...]*

- We would like to emphasize that the study was intended to understand how damage accumulates in curling stones, and by extension, more generally in granitoids. Curling stones as study objects were not simply a convenient opportunity to understand damage in granitoids and certainly not a gimmick. We intend to include a sentence on l. 41 of the preprint to make our intention clearer: "In this contribution, we seek to elucidate how damage accumulates in curling stones, and, by extension, more generally in granitoids." We trust that this specificity focuses the introduction.

*[...] The main issues with the manuscript are 1) the oversimplified mechanical analysis of the problem, 2) the applicability of the experiments to natural deformation processes and the insufficient embedding of this study within existing (experimental) studies. More on these main issues follows next, but they are also often at the basis of the more detailed comments given further below. Since these issues are key to the conclusions of the paper, I recommend major revisions.*

- We acknowledge the key limitations that have been raised with respect to the simplistic mechanical analysis, as well as the limitations in literature included on natural deformation processes and other experimental studies. However, we would like to re-iterate that the primary intent of this study was focused on understanding how damage accumulates in curling stones, rather than comparing curling stones to other experimental studies.

- Our simple mechanical analysis represents a first-order approximation of the stresses and strains involved in curling stone impacts. The Reviewer acknowledges that determining dynamic stresses is challenging and the modelling required would be beyond the scope of this contribution. We intend to explicitly describe these limitations and areas of future investigation in the *Conclusions* section: "This work has several limitations that could be expanded through future work. Most critically, the approaches used to estimate the stress and strain of the impacts are oversimplified and do not take into account that stress propagates as waves at the p-wave velocity of the material. This leads to stress, strain, and strain-rate fields being heterogeneous and evolving with time and space. Future efforts in modelling the dynamic stresses will be able to address the potential for heterogeneous, evolving stresses as a possible mechanism for damage. Along these lines, experiments on the dynamic strength of curling stones would better place curling stone impacts within the context of natural deformation processes. [...]".

- We will address the two overarching comments in greater detail below, in reference to the specific comments that have been raised.

***The mechanical analysis (section 2):*** *Averages for stress, strain, and strain rate over the duration of loading are presented first, following impact mechanics. A second measure, the maximum stress, is based on contact mechanics of two elastic bodies. Both are insufficient for these experiments:*

- *First, they do not take into account that stress at such high loading rates/impacts now propagate as waves. The stress, strain, and strain rate fields vary highly in space and time. The paper attempts to explain the observation of damage concentrated near the impact, and heterogeneous stresses that decay away from the impact may be a key component that is overlooked by the authors. Understanding this and achieving "stress equilibrium" over an entire sample (i.e., stress is the same everywhere, but still varies over time) is already a challenge in typical simple uniaxial experiments on Split-Hopkinson Pressure Bars (SHPSs) performed at comparable strain rates (e.g., Nemat-Nasser et al., 1991; Zhang and Zhao, 2013; Aben et al., 2017), but a necessary one to say something about representative dynamic material parameters or constitutive dynamic behaviour. The impact mechanics approach uses a time average, which may provide a first-order rough sample-wide estimate, but does not provide the level of detail to match with the microstructures. The contact mechanics approach entirely lacks any consideration for the dynamic nature of the experiments.*
    - Thank you for raising this issue. In the *Mechanical Background* section (l. 73 of the preprint), we intend to include a paragraph mentioning the challenge of determining the parameters of dynamic deformation, as well as the limitations of the impact and contact mechanics approaches: "As a note, curling stone impacts are inherently dynamic in nature. As such, stress is expected to propagate in waves, and the distribution, stress, and strain rate is expected to be heterogeneous over space and time. Achieving stress equilibrium in simple, dynamic uniaxial experiments is already challenging (e.g., split-Hopkinson pressure bar experiments; Nemat-Nasser et al., 1991; Zhang and Zhao, 2014; Aben et al., 2017), and thus the style of damage produced by curling stones is expected to differ from dynamic uniaxial experiments due to the effects of heterogeneous stress, strain, and strain rate. The impact mechanics approach uses a time average and provides a first-order estimate of the stress, strain, and strain rate of curling stone impacts. On the other hand, the contact mechanics approach does not consider the dynamic nature of the experiments, but it does consider stress heterogeneities resulting from the geometry of the collisions. The complexity in determining the evolution of stress, strain, and strain rate cannot be fully documented in the on-ice experiments; as such, the reader is cautioned regarding the interpretation of the overly simplistic stress, strain, and strain-rate values determined in this contribution."
    - Although we agree that the contact mechanics approach ignores the dynamic nature of the experiments, we argue that the geometry of the collision (i.e., Hertzian contacts, which may also form due to dynamic collisional processes) has a greater impact on the heterogeneous stress distribution of the impacts and thus leads to the development of crescentshaped fractures. Specifically, we will add a paragraph at l. 379 of the preprint addressing this, along with a corresponding figure from Padture and Lawn (1995) and Lawn (1998) illustrating these similarities (Fig. AC1, below): "All of these observations closely resemble the damage observed in cyclic Hertzian indentation experiments on silicon carbide ceramics with homogeneous grain sizes (Fig. [AC1]; Padture and Lawn, 1995), where (1) an initial cone fracture forms during the first indentation cycle, then (2) a damage zone develops in the collet zone over 100 to 10,000 cycles, and (3) material is ejected from the collet zone over 10,000 to 1,000,000 cycles (cf. exfoliation in the collet zone of crescent-shaped fractures). Similar to our observations, it is shown that damage is largely restricted to the Hertzian cone fractures and their associated collet zones, with virtually no damage outside of these areas.[...] In summary, almost all of the damage features visible in curling stone impacts can be attributed to cyclic loading of Hertzian contacts, and this forms the basis for our conceptual model for damage evolution in curling stones."

$n = 10^0$

$n = 10^2$

$n = 10^4$

$n = 10^6$

500 μm

- ○ Fig. AC1. Damage evolution of silicon carbide ceramics with homogeneous grain sizes under cyclical spherical indentation (radius = 3.18 mm, force = 1000 N, frequency = 10 Hz) under Nomarski interference, n = number of loading cycles. From Padture and Lawn (1995); Lawn (1998).
- ○ We have additionally included limitations in the *Conclusions* section to guide future work in this area (see previous comment).

- *Second, the average strain from the impact mechanics approach and the entire contact mechanics approach are based on the assumption of linear elastic behaviour of the material. The microstructural study reveals this is not the case, and inelastic deformation (most likely at higher stresses) is accumulated. This affects the stress-strain relation and the assumption of linear elasticity leads to an overestimation of the actual stresses at the contact. How this combines with the dynamics at the contacts previously mentioned is unclear, and so the calculated maximum stress estimates are unfounded.*
    - ○ While it is true that the accumulation of fractures leads to the interpretation that inelastic deformation is accumulated, our first-order energy conservation calculations on ll. 310–321 indicate that an overwhelming majority of the energy (~ 96 %) is recovered by the acceleration of the resting stone or can be accounted for by friction. This means that the majority of deformation is likely to take place within the elastic region of the stress-strain curve.
    - ○ Additionally, it must be recognized that the macroscopic fractures develop over a span of hundreds to thousands of impacts. Assuming a curling stone receives 4 impacts per game (a conservative estimate – the maximum would be 16 impacts per game), 4 games per day, 30 days per month, and 6 months in a given season, a curling stone would receive roughly 2,900 impacts per season or 29,000 impacts in a 10-year period. Given that the circumference of a curling stone is around 90 cm, this means that each centimetre of the stone would receive, at minimum, around 320 impacts in that 10-year period. In other words, the damage that each impact contributes to a curling stone is expected to be minor (as curling stones rarely, if ever, experience catastrophic failure). These observations are consistent with the majority of deformation occurring within the elastic region. Note that this estimate is subject to several variables, including: (1) the number of games played per day, which is dependent on membership of a given facility; (2) the length of the season; and (3) whether the stones are consistently thrown in the same order or re-arranged, as stones delivered earlier in an end may have a greater probability of receiving impacts than those delivered late in an end. All of these variables change depending on location.
    - ○ We would like to clarify that no maximum stress estimates are provided for the curling stone impacts. Lines 189–193 clearly indicate that the values represent ***average*** stress estimates for the maximum impact velocity scenario of 2.93 ms$^{-1}$. To avoid confusion, we will additionally remove the

equations related to the "maximum Hertzian stress" in the *Mechanical Background* (Equations 5–7 of the preprint), as these can be integrated directly into the average Hertzian stress calculations.

- ○ We intend to include a statement after l. 60 of the preprint mentioning considerations regarding the assumption of elastic deformation: "It is important to note that the stress, strain, and strain-rate approximations introduced by the impact mechanics approach assume that elastic deformation occurs. However, the development of damage in curling stones indicates that some degree of inelastic deformation is present, meaning that the assumption of elastic deformation may oversimplify the calculations. However, given that most of the energy of curling stone impacts goes into the motion of the stationary stone (see *Discussion*), and that the damage occurs over hundreds to thousands of impacts ([insert reference to this reply]), it is inferred that the majority of deformation occurs within the elastic region."

*Finally, it may be beyond the presented study to measure and/or model the full stress field during the impact loading, as it will be hard to measure on a large sample with a complex geometry. However, the issue of dynamic loading and wave propagation should be addressed nonetheless so that readers can place the averaged stress/strain/strain rate estimates in the correct context, and it may aid to explain the microstructural results. I recommend to remove the maximum stress estimates all together.*

- We agree that the measurement and/or modelling of the full stress field during impact loading is out of the scope of the present study, although it is an extremely intriguing future endeavor. This will be addressed in the *Conclusions* of the manuscript (as mentioned above).
- We will include the issue of dynamic loading and wave propagation in the *Mechanical Background* section (see insertion on l. 73 in the above comments), as this is the most suitable place to point out the over-simplifications involved in calculating the stress and strain.
- We reiterate that the stress estimates presented are average estimates, not maximum estimates (ll. 189–193). The word "maximum" refers to the highest impact velocities achieved in the experiments. As such, the estimates remain in the manuscript.

*Scientific context of the study: The previous major comment is partially related to the issue of sub-par scientific context given in the manuscript. Starting with the introduction, the scientific aim remains elusive and vague. What is specifically the question that needs to be answered regarding repeated impact loading? What is the current state of knowledge? Very little literature is cited on the subject of dynamic loading (I have provided a few references throughout and at the end of this review), let alone on the effects of repeated dynamic loading (e.g., Doan & d'Hour (2012), Aben et al (2016); Braunagel & Griffith 2019). A decent set of literature to set up the specific problem to be solved in this manuscript is not only helpful for the reader, but would also have highlighted the difficulties in performing and analysing dynamic loading experiments to*

*the authors (see first main issue). On the issue of applicability of the impact experiments to natural deformation processes, the comparisons with stresses and strain rates are too hasty and incorrect (see detailed comments on them below). One field (which happens to be my expertise) where indeed the experiments could be analogues are dynamic loading in fault damage zones (e.g., Doan & Gary 2009; Yuan 2011; Braunagel & Griffith 2019). Here, the idea of a damage zone "protecting" undamaged regions has also been opted (Ostermeijer et al., 2022). This has been overlooked by the authors, which makes me worry about other potentially overlooked but relevant literature. I hope that a substantial effort on embedding this study within existing literature can help to mature the paper considerably, as it now feels to much as a fun "gimmick" with curling stones where the application was an afterthought.*

- We again reiterate that the primary goal of this study was to understand how damage accumulates in curling stones, and by extension, more generally in granitoids. Curling stones were not simply a convenient opportunity to understand damage in granitoids and certainly not a gimmick.
- To improve the issue of dynamic, cyclic loading, we intend to include points in the *Mechanical Background* to illustrate the limits of the stress, strain, and strain rate calculations, as well as highlight the challenges with performing and analysing dynamic loading experiments (see above comments regarding insertion on l. 73). We will also add points in the *Conclusions* (see above comments) to highlight the limitations and future directions with respect to modelling the dynamic deformation of curling stone materials.
- We will include the Ostermeijer et al. (2022) reference on l. 375 of the preprint: "Additionally, findings from Ostermeijer et al. (2022) further support the concept that damage zones can dissipate energy and protect undamaged regions from accumulating damage." Thank you for this addition.

**Detailed comments**
*Line 63: Double meaning for the notation a (acceleration and semi-major axis).*
- The typeset of α (acceleration) and a (semi-major axis) were differentiated in the text. However, given the potential confusion, we have opted to remove the reference to acceleration from Equation 2 of the preprint, as it is not critical to the derivation.

*Line 156: "Higher probability thresholds… in some tiles.": Why is this an issue? At least, with a 90% threshold, the remaining 90% of pixels has been robustly identified, which is not the case when using a low threshold of 50% probability. The aim of the identification exercise is to identify the components of the rock with a certain amount of confidence, not to identify everything by basically just guessing.*
- Due to partial surface/volume interaction effects, it is very challenging to robustly segment minerals which display extensive solid solution and porosity from backscattered electron imaging (BSE) data of rock-forming silicate minerals. In our case, the difficulty stems mainly from the alkali feldspar, which (1) displays perthitic textures between albite and K-feldspar, leading to considerable BSE brightness variation; (2) the porous nature of the feldspars, leading to partial

surface effects; and (3) that albite has a low BSE response, leading to the mineral being misidentified as quartz where partial volume effects are in play.

- To illustrate these effects visually, we have subdivided Fig. 11a in the main text into 4 x 4 subdivisions (columns A–D, rows 1–4). Evaluating subsection A1 from a visual perspective (Fig. AC2a, below), a probability threshold of 50 % (Fig. AC2b) successfully segments alkali feldspar domains, whereas a probability threshold of 90 % (Fig. AC2c) leads to an excessive proportion of unassigned pixels. It can also be seen that the cleavages in the arfvedsonite lead to a much higher degree of unassigned pixels due to partial surface effects. In general, quartz suffers the least from increasing the probability threshold, given that it does not show substantial solid solution, and it contains few pores and fractures. As such, increasing the probability threshold to 90 % leads to substantial biases in segmentation of minerals, which also biases the modal mineralogical data and inferences based on that data.

[Figure]

-
- Fig. AC2: Visual segmentation analysis of a subsection A1 of Fig. 11 from the main text: (a) BSE image; (b) segmentation based on a probability threshold ($p$) of 50 %; (c) segmentation based on $p$ = 90 %.
- To illustrate the effect that increasing probability thresholds has on relative abundances of the minerals, we also analyzed the proportion of pixels assigned to each mineral in three different subsections of Fig. 11 from the main text (subsections A1, B3, and C4; see Fig. AC3 below).

[Figure]

-
- Fig. AC3: Proportions (left) and normalized proportions (right) for segmentations of three different subsections (A1, B3, and C4).
- In Fig. AC3, it can be seen that the number of unclassified pixels for a 50 % probability threshold ranges from 5–8 %, whereas that for a 90 % threshold yields 50–60 % unclassified pixels. The increase in unclassified pixels increases substantially from a probability threshold beyond 50 %, mainly coupled with a decrease in the proportion of pixels assigned to alkali feldspar. The inflection point where alkali feldspar classification becomes poor appears to be around a

probability threshold of 50–60 % depending on the subsection analyzed. As such, a probability threshold of 50 % is appropriate.
- These figures will be added to File S3 in the supplement to further strengthen the arguments for a 50 % probability threshold.

*Line 171: Why were the other 21 experiments discarded/not reported here?*
- A significant component of conducting experimental science involves getting the experiment to work successfully, and then optimizing data acquisition parameters. Also, even with a world champion delivering the curling stones, there is always an element of human error present, both in the execution and recording of the experiments. These aspects have ultimately impacted the selection of which data to use.
- The 30 experiments we conducted in total consist of (a) experiments with Fujifilm Prescale HHS pressure-sensitive films (n = 4); (b) preliminary tests of aluminium foils without velocity analysis (n = 4); (c) aluminium foil tests with velocity analysis (n = 9); and (d) high-speed camera experiments (n = 13). The discarded tests are described in greater detail as follows:
    - (a) Fujifilm Prescale HHS pressure-sensitive films: 3/4 experiments were discarded due to human errors during testing of parameters: velocity not measured (n = 1); GoPro frame rate was set too low, leading to large errors in velocity estimation (n = 1); and incoming stone partially missed pressure-sensitive film (n = 1).
    - (b) Preliminary tests of aluminium foils: 3/4 experiments were discarded due to method development (velocity was not measured); Exp. 0.4 was kept as it approximates the contact between two touching stones.
    - (c) Aluminium foil tests with velocity analysis: 3/9 experiments were discarded due to representing exploratory experiments of two touching rocks hit by a third rock, representing an impact configuration that was ultimately beyond the scope of the present work.
    - (d) High-speed camera experiments: 12/13 experiments were discarded. Eleven of these experiments used a lower frame rate than desired and at impact velocities lower than the maximum-velocity scenario (10-30 kfps). Of the two remaining experiments conducted at 40 kfps at maximum velocity, one experiment was discarded because the collision occurred partially outside of the field of view.
- On l. 82 of the preprint, we will itemize which experiments correspond to which analyses: "These 30 experiments consisted of (a) experiments using Fujifilm Prescale HHS (n = 4); (b) preliminary tests of aluminium foils without velocity analysis (n = 4); aluminium foil tests with velocity analysis (n = 9); and (d) high-speed camera experiments (n = 13)." On l. 84 of the preprint, we will also include a reference to this reply in the discussion regarding the nature of the discarded experiments.

*304-325: This whole section is full of statements without proper arguments or references to support them. In more detail:*

- For clarity, this whole section will be rewritten as follows: "Our study explores a particular configuration of cyclical loading in granitoids: hundreds to thousands of impacts with dynamic loading of Hertzian contacts, reaching dynamic stress magnitudes that exceed the quasi-static compressive strengths of the material. This configuration differs from conventional, quasi-static loading and fatigue tests (e.g., Cerfontaine and Collin, 2018), given the higher strain rates and the geometric nature of the Hertzian contacts. Although the strain rates more closely match cyclic dynamic loading tests (i.e., split-Hopkinson pressure bar tests; Aben et al., 2016; Doan and d'Hour, 2012; Braunagel and Griffith, 2019), the configuration differs by (1) the geometry of the contact surfaces (curved versus flat contacts) and resultant heterogeneous stress field evolution; as well as (2) the fact that most of the energy is invested into the unbounded acceleration of the resting stone (see below for discussion), in contrast to dynamic loading experiments, in which the sample is bounded by two bars (the incident and transmitted bars). Nevertheless, the case of curling stone collisions is peculiar in that the stones support these impacts for much longer than cyclical loading tests would suggest.

- "Determining the impact energy that is invested into the acceleration of the resting stone requires the exact quantification of the initial and final velocities of the moving and stationary stones. Although these were tracked during each experiment using the GoPro cameras, calculations of kinetic energy using these data yielded values of low precision. As a first-order approximation, the high-speed camera experiment no. 2.13 (stone mass 18.41 kg) yielded pre-impact velocities (and kinetic energies in parentheses) of 2.74 ms$^{-1}$ (69 J) and 0 ms$^{-1}$ (0 J) for the incoming and stationary stones, respectively. Immediately after the impact, these changed to 0.09 ms$^{-1}$ (<< 1 J) and 2.66 ms$^{-1}$ (65 J) for the incoming and now-accelerated stone. This implies an energy loss of ~ 4 J (~ 6 %). This energy loss estimate also includes work done by friction while the stones slide on the ice surface, which represents ~ 1 J when applying a coefficient of friction of 0.1 (Nyberg et al., 2013), meaning that ~ 3 J or ~ 4 % of the impact energy is dissipated in the stone. In the context of natural phenomena such as earthquakes, estimates of energy dissipation by fracturing vary from 1 % to > 50 % (Rockwell et al., 2009; Wilson et al., 2005). At similar strain rates, cyclic split-Hopkinson pressure bar experiments by Xu et al. (2024) recorded an energy absorption between 15–25 %, which is much higher than the 4 % calculated for curling stone impacts in this work. These calculations emphasize the dominantly elastic nature of the deformation, where most of the energy is transferred to unbounded motion of the stationary stone, as opposed to producing damage within the rock. Our data indicate that the remaining energy that is not transferred to the motion of the stationary stone is dissipated by propagating crescent-shaped fractures and localizing damage within their collet zones, as well as by ejecting rock powder at the surface of the curling stones."

- We will now describe how these proposed changes address the more detailed comments below.

- *Line 305: Why potentially?*

- ○ Thank you for raising this issue. We will delete the word "potentially" on line 305 of the preprint.

- *Line 307-309: What is the evidence for this statement?*
  - ○ This paragraph will be rephrased to show that ~ 96 % of the energy is transferred to motion of the stationary stone or friction, as opposed to producing damage within the rock, which is a key difference to cyclic dynamic loading tests, where the sample is sandwiched by incident and transmitted bars (and where the absorbed energy is much higher for comparable strain rates, 15–25 %; Xu et al., 2024).

- *Line 317: Who does the 4% compare with other studies on energy dissipation by fracturing during geo-dynamic events such as earthquakes?*
  - ○ Estimates for energy dissipation during earthquakes vary considerably from 1 % to > 50 % (Rockwell et al., 2009; Wilson et al., 2005). We will additionally add a reference noting that the energy absorption of split-Hopkinson pressure bars ranges between 15–25 % for comparable strain rates (Xu et al., 2024), which is much higher than the 4 % for curling stone impacts.

- *Line 318-319: I do not agree with/do not understand this sentence, what exactly is significant (possibly use another word here, as it is inherently related to statistical metrics)? The sentence implies that the calculations are a surprising outcome relative to a hypothesis that is never stated. Aside, the stresses are not instantaneously (they are "transferred" from stone to stone by stress waves at the P-wave speed of the material). What is meant by the stresses are recovered by the acceleration of the resting stone?*
  - ○ We will modify this sentence to read: "These calculations emphasize the dominantly elastic nature of curling stone impacts, where most of the energy is transferred to unbounded motion of the stationary stone, as opposed to producing damage within the rock." We will remove the instantaneous wording, and the phrase "stresses are recovered by the acceleration of the resting stone."

- *Line 320-321: This sentence contains a lot unclarity: What does "efficient in dissipating energy" mean? That the rock dissipates a lot of the impact energy, or that it does not dissipate a lot? "once it is saturated in fractures": Where are the numbers to back this up, does the 4% mentioned earlier change with number of impacts?*
  - ○ We will modify this sentence to read: "Our data indicate that the remaining energy that is not transferred to the motion of the stationary stone is dissipated by propagating crescent-shaped fractures and localizing damage within their collet zones, as well as by ejecting rock powder at the surface of the curling stones." This avoids the ambiguous terminology of "efficient in dissipating energy" and focuses purely on the style of damage accumulation in curling stones as a result of the damage characterization.

- ○ As we have not studied the full lifetime of impacts of a curling stone, we do not know if the 4 % energy absorption changes with the number of impacts. It must also be emphasized that this absorption estimate is only a first-order estimate because the uncertainties involved are high.

*Line 333-334: The sentence damage must exceed a threshold to produce fatigue damage is vague and needs more explanation. Is this conditional on repeated loadings at the same stress level? Clarify that Zhou et al (2018) performed high strain rate (i.e., dynamic) loading experiments. Are the strain rates comparable? Similar, but earlier, work on repeated dynamic loading in compression was done by Doan & d'Hour (2012), Aben et al (2016)).*

- We will modify this sentence to include that the strain rates used in Zhou et al. (2018) are comparable to curling stone impacts (22 s$^{-1}$). We will also specify that the critical fatigue stress is half of the dynamic rock strength, and that the condition tested was for repeated loadings at the same stress level.
- We will also add a clarification that dynamic rock strength can decrease by nearly twofold by cyclic loading, the implication being that high-velocity impacts may initiate damage in curling stones, but lower-velocity impacts may still contribute to damage accumulation.
- We will also add a note specifying that the dynamic rock strength of curling stone impacts would be expected to increase beyond the quasi-static strength at higher strain rates.
- The proposed modified text will read as: "1. Our calculations from on-ice experiments indicate that the dynamic stress magnitudes of the impacts of curling stones (300–680 MPa) exceed the quasi-static uniaxial compressive strength ranges recorded for curling stones (230–520 MPa; Fig. 13; Nichol, 2001; Leung, 2020). Our post-mortem investigation indicates that these impact stresses are responsible for the observed damage. The observation of the dynamic stress exceeding the quasi-static strength of the stones may lead to the incorrect assumption that curling stones should experience failure on one impact. This is because rock strength increases with increasing strain rate, and at the dynamic strain rates of curling stone impacts ($\sim 10^1$ s$^{-1}$), the dynamic strengths of granitoids are typically 1.5–1.9 times higher than their quasi-static strengths (Braunagel and Griffith, 2019; Doan and d'Hour, 2012; Li et al., 2000; Green and Perkins, 1969; Hokka et al., 2016). This implies that the dynamic stresses of curling stones impacts may not actually exceed the expected dynamic strength of the rocks. However, these dynamic stresses likely exceed the threshold required for fatigue damage for repeated loadings at the same stress level (about half the maximum dynamic compressive strength of a single impact; Zhou et al., 2018), thus explaining why damage accumulates in curling stones. It is also important to highlight that dynamic rock strength decreases nearly twofold with cyclic loading (Braunagel and Griffith, 2019; Doan and d'Hour, 2012). This means that once curling stones are damaged by high-velocity impacts, lower-velocity impacts may also contribute to the damage accumulation."

*Paragraph starting on Line 335 and table 1: This comparison exercise with stress magnitudes of natural deformation processes is too simplistic and leads to erroneous analogies between processes: Stress is a tensorial quantity, the curling experiments performed here are by approximation uniaxial so that what is described as "the" stress is one of the diagonal uniaxial stress component. This cannot be compared to earthquake stress drops, which are shear stresses, or to lithostatic stresses which are meaningless without knowing the other principal stress magnitudes. It is unclear how thermal and mining-induced stresses are defined (are they stress invariants or uniaxial stress components?). In short, from this table, the only directly comparable ones are rockfalls and ballistic impacts.*

- We will include footnotes under Table 1 explaining that mining-induced stresses and thermal stresses represent deviatoric stresses (with adjusted values corresponding to deviatoric stresses), whereas lithostatic stresses are stress invariants and are only broadly comparable to curling stone impacts. We will also change wording in the main text (on ll. 338–339 of the preprint) specifying that curling stone impacts "most closely resemble rockfall stresses."

*Line 378: I guess the authors mean that, rather than dampening damage (if such a thing is possible), the stresses are dampened by energy dissipation in the damage zone.*

- Thank you for suggesting this change in wording. This paragraph will be integrated with the previous paragraph and be rewritten as: "That is, the stresses are dampened by energy dissipation in the damage zone by utilizing pre-existing structures, rather than forming new ones."

*Table 2: Similar comment as to table 1, be cautious when comparing strain rates with other strain rates. Comparing uniaxial strain rates with coseismic slip rates is incorrect, as the latter is a shear strain rate. There is plenty of literature on high strain rate deformation experiments at uniaxial conditions in compression that should be used to put this work into context.*

- Thank you for this comment. We will replace the co-seismic slip rates with co-seismic rock pulverization estimates (uniaxial strain rates) inferred through split-Hopkinson pressure bar estimates of damaged rocks for Table 2. We will also include commentary in the main text on ll. 345–350 of the preprint.
- The proposed modified text on ll. 345–350 is: "3. The magnitude of strain rates in curling stone impacts ($24 \pm 4$ s$^{-1}$) can be classified as a high strain rate response (Zhang and Zhao, 2014) and suggests that there is a significant dynamic component involved in curling stone impacts. This is corroborated by the ejection of rock powder from the striking bands after impacts between curling stones, as observed by high-speed camera footage. Additionally, the presence of potential spalling microcracks in our post-mortem analysis also supports this interpretation. In terms of strain rate, the closest analogue to curling stone impacts is co-seismic rock pulverization as investigated by cyclic split-Hopkinson pressure bar experiments. These experiments show that the style of failure changes from single fractures to multiple fragments (pulverization) above 85–150 s$^{-1}$ for previously damaged samples (Doan and d'Hour, 2012; Aben et al., 2016). The lower average elastic strain rate of curling stone impacts calculated by this

study generally supports the lack of significant pulverization in curling stones. However, it must be considered that stress and strain are heterogeneous, and their tensorial fields evolve over time and space in curling stone impacts, with stress and strain decaying with distance into the striking band. Thus, the minor amount of pulverized rock powder and spalling microcracks near the surface of the striking band could be related to high local strain rates at the contact surface, although they may also be a product of free-surface effects or associated with damage zones associated with Hertzian contacts (see following subsection; Padture and Lawn, 1995). In general, the fact that damage originates from the surface of striking bands rather than the center of the curling stone (as in split-Hopkinson pressure bar experiments) indicates that free-surface effects and their associated, heterogeneous stress and strain distributions are expected to have a greater impact on the damage geometry compared to their high strain-rate natures. Nevertheless, our microstructural observations might provide insights into the damage response of seismically loaded granitoids, including pulverization."

*Line 361: It is confusing to encounter stress component notation whilst the rest of the manuscript only speaks of a peak and average stress. How does this stress component relate to the stresses derived from the experiments? Clarify that this specific stress component mentioned here is the maximum tensile stress component in the vicinity of the Hertzian contact.*

- Thank you for raising this issue. We will remove the stress component notation and specify that the stress refers to the maximum compressive stress component, which is equivalent to the main stress component derived from the experiments.
- The proposed modification to l. 361 is: "The conoid 3D geometry of Hertzian cone fractures as displayed by crescent-shaped fractures follows a path that maximizes the strain energy release during the impact (Kocer and Collins, 1998) and is broadly parallel to the stress trajectories of the maximum compressive stress component (Frank and Lawn, 1967), which correlates to the stress component measured in the on-ice experiments. [...]"

*Figure 14 is not referenced in the main text.*

- The subfigures associated with Fig. 14 were previously referenced in the preprint, but we will add an explicit reference to Fig. 14 on l. 355 of the preprint.
- We noticed that Fig. 13 was not referenced and will add a reference on l. 332: "[...] exceed the quasi-static uniaxial compressive strength ranges recorded for curling stones (230–520 MPa; Fig. 13; Nichol, 2001; Leung, 2020)."

*References*
*Nemat-Nasser, S., J. B. Isaacs, and J. E. Starrett (1991), Hopkinson techniques for dynamic recovery experiments, Proc. R. Soc. London, Ser. A, 435(1894), 371–391.*
*Zhang, Q. B., and J. Zhao (2013), A review of dynamic experimental techniques and mechanical behaviour of rock materials, Rock Mech. Rock Eng., doi:10.1007/ s00603-013-0463-y*

*Aben, F. M., Doan, M.-L., Gratier, J.-P., & Renard, F. (2017). Coseismic damage generation and pulverization in fault zones: Insights from dynamic Split-Hopkinson Pressure Bar experiments. In M. Y. Thomas, H. S. Bhat, & T. M. Mitchell (Eds.), Evolution of fault zone properties and dynamic processes during seismic rupture (pp. 47–80). Washington, DC: John Wiley & Sons*

*Doan, Mai-Linh, and Virginie d'Hour. "Effect of initial damage on rock pulverization along faults." Journal of Structural Geology 45 (2012): 113-124.*

*Aben, F. M., et al. "Dynamic fracturing by successive coseismic loadings leads to pulverization in active fault zones." Journal of Geophysical Research: Solid Earth 121.4 (2016): 2338-2360.*

*Xia, Kaiwen, and Wei Yao. "Dynamic rock tests using split Hopkinson (Kolsky) bar system–A review." Journal of Rock Mechanics and Geotechnical Engineering 7.1 (2015): 27-59.*

*Doan, Mai-Linh, and Gérard Gary. "Rock pulverization at high strain rate near the San Andreas fault." Nature Geoscience 2.10 (2009): 709-712.*

*Yuan, Fuping, Vikas Prakash, and Terry Tullis. "Origin of pulverized rocks during earthquake fault rupture." Journal of Geophysical Research: Solid Earth 116.B6 (2011).*

*Braunagel, Michael J., and W. Ashley Griffith. "The effect of dynamic stress cycling on the compressive strength of rocks." Geophysical Research Letters 46.12 (2019): 6479-6486.*

*Ostermeijer, Giles A., et al. "Evolution of co-seismic off-fault damage towards pulverisation." Earth and planetary science letters 579 (2022): 117353.*

- Thank you for providing these references. These will improve the scientific context and relevance of the manuscript to natural deformation processes.

References cited in this reply:

- Frank, F. C. and Lawn, B. R.: On the Theory of Hertzian Fracture, Proceedings of the Royal Society of London. Series A, Mathematical and Physical Sciences, 299, 291–306, 1967.
- Green, S. J. and Perkins, R. D.: Uniaxial Compression Tests At Varying Strain Rates On Three Geologic Materials, The 10th U.S. Symposium on Rock Mechanics (USRMS), 1968.
- Hokka, M., Black, J., Tkalich, D., Fourmeau, M., Kane, A., Hoang, N.-H., Li, C. C., Chen, W. W., and Kuokkala, V.-T.: Effects of strain rate and confining pressure on the compressive behavior of Kuru granite, International Journal of Impact Engineering, 91, 183–193, https://doi.org/10.1016/j.ijimpeng.2016.01.010, 2016.
- Kaiser, P. K., Yazici, S., and Maloney, S.: Mining-induced stress change and consequences of stress path on excavation stability — a case study, International Journal of Rock Mechanics and Mining Sciences, 38, 167–180, https://doi.org/10.1016/S1365-1609(00)00038-1, 2001.
- Lawn, B. R.: Indentation of Ceramics with Spheres: A Century after Hertz, Journal of the American Ceramic Society, 81, 1977–1994, https://doi.org/10.1111/j.1151-2916.1998.tb02580.x, 1998.
- Li, H. B., Zhao, J., and Li, T. J.: Micromechanical modelling of the mechanical properties of a granite under dynamic uniaxial compressive loads, International

Journal of Rock Mechanics and Mining Sciences, 37, 923–935, https://doi.org/10.1016/S1365-1609(00)00025-3, 2000.

- Padture, N. P. and Lawn, B. R.: Contact Fatigue of a Silicon Carbide with a Heterogeneous Grain Structure, Journal of the American Ceramic Society, 78, 1431–1438, https://doi.org/10.1111/j.1151-2916.1995.tb08834.x, 1995.
- Rockwell, T., Sisk, M., Girty, G., Dor, O., Wechsler, N., and Ben-Zion, Y.: Chemical and Physical Characteristics of Pulverized Tejon Lookout Granite Adjacent to the San Andreas and Garlock Faults: Implications for Earthquake Physics, in: Mechanics, Structure and Evolution of Fault Zones, edited by: Ben-Zion, Y. and Sammis, C., Birkhäuser, Basel, 1725–1746, https://doi.org/10.1007/978-3-0346-0138-2_9, 2010.
- Wilson, B., Dewers, T., Reches, Z., and Brune, J.: Particle size and energetics of gouge from earthquake rupture zones, Nature, 434, 749–752, https://doi.org/10.1038/nature03433, 2005.
- Xu, H., Rong, C., Wang, B., Zhang, Q., Shen, Z., and Jin, Y.: Dynamic mechanical behavior and energy dissipation characteristics of low-temperature saturated granite under cyclic impact loading, Sci Rep, 14, 26840, https://doi.org/10.1038/s41598-024-74059-3, 2024.

---

## Author Comment (AC2)

**Reviewer 2 (Dr. Charalampidou)**

*[...]A general observation relates to the clarity around the materials studied. It's not always evident which results correspond to Ailsa Craig Common Green and which to Ailsa Craig Blue Hone. While this distinction is made in some sections of the manuscript, it's missing in others, which makes interpretation more difficult. Additionally, it's not consistently clear whether the two rock types exhibit similar or different mechanical behaviours across the various analyses - especially since different types of results are presented for each (with the exception of a couple of graphs).*

- In the *Materials and Methods* section, we will insert several sentences beginning on ll. 79 of the preprint summarizing which rock types are used for which areas of the study (on-ice experiments, macroscopic damage characterization, and microscopic damage characterization). We will also describe the rocks from Trefor for context. This should help to specify which rocks are used in which aspects of the study. We will additionally include the rock types studied for each figure to reinforce this point.
- The modified paragraph on ll. 75–79 will read as: "We studied curling stones from both Ailsa Craig (Firth of Clyde, Scotland) and Trefor (Llŷn Peninsula, North Wales). Two types of rocks from Ailsa Craig are used in curling stones: Ailsa Craig Common Green, which is currently used for the striking bands of Olympic-standard curling stones; and Ailsa Craig Blue Hone, which was used as the striking band in older stones, but is currently inserted into the running bands of the stones (Leung and McDonald, 2022; see Fig. 1b for locations of the running band and striking band). Two types of rocks from Trefor are used in curling stones: Blue Trefor and Red Trefor, both of which are typically used for striking bands. In general, this study focuses on the rocks from Ailsa Craig, although we integrated several different rock types into the study: the on-ice experiments used Ailsa Craig Common Green; the macroscopic damage characterization predominantly used Ailsa Craig Blue Hone, with ancillary Ailsa Craig Common Green and Red Trefor samples; and the microfracture descriptions used only Ailsa Craig Blue Hone. The on-ice experiments were limited by the availability of actively used rocks at Curl Edinburgh. Different rock types were used in the macroscopic damage characterization to illustrate the full extent of damage in curling stones. Lastly, Ailsa Craig Blue Hone displayed the most developed crescent-shaped fractures and was thus the focus for the microstructural aspects of the study."
- In a general sense, the specific rocks used in the on-ice experiments are less critical, as the goal of these experiments is to document the stresses and strains of the impacts. The contact mechanics calculations may vary based on the choice of Young's modulus (here, we used experimental data for Ailsa Craig Common Green from Leung 2020), but the calculations are approximate and thus the substitution of a different Young's modulus would not be expected to substantially change the outcome. For reference, the experimental Young's modulus determined by Leung (2020) for Ailsa Craig Common Green is 39 GPa, Ailsa Craig Blue Hone 37 GPa, and Red Trefor 29 GPa (Blue Trefor was not measured due to issues encountered during the experimental process). With

regard to the Ailsa Craig suite, the Young's moduli of the Ailsa Craig varieties are considered to be equivalent. For Red Trefor, substituting a Young's modulus of 29 GPa for 39 GPa yields a mean stress of 500 MPa for the maximum-velocity scenario (cf. 680 MPa for 39 GPa). Although this value is somewhat lower, it still lies at the maximum end of the uniaxial compressive strength for the rocks. Note that this calculation is only meant to be conceptual, as different rock types may experience different degrees of compression and hence develop differing contact areas, and this is considered beyond the scope of the present contribution.

- The greatest limitation in interpretation comes from the use of different rocks in the macroscopic damage analysis (in particular, Figs. 8-9). It was unfortunately very challenging to obtain striking band samples of the same rock type with varying damage states, and we do not know for how long these stones were played. As such, we chose to limit discussion on comparing between the rock types and their mechanical behaviors, instead focusing on the distribution of crescent-shaped fractures among the sample suite. In the macroscopic damage analysis results, the rock types are clearly labelled on Figs. 8-9. We will additionally add a paragraph after l. 239 specifying the limitations of interpreting the effect that different mechanical behaviors may have on the development of crescent-shaped fractures: "As a note, our macroscale analysis of the morphology and distribution of crescent-shaped fractures in this section utilizes samples spanning several different rock types. Due to the lack of sample availability, it was not possible to compare the same rock type with different damage states. As such, we do not consider the effect that different mechanical properties may have on the development of crescent-shaped fractures."
- To focus on how microfractures evolve, and to implicitly avoid issues with mixing and matching between different rock types, the microscale damage analysis uses only Ailsa Craig Blue Hone samples.
- In the *Conclusions*, we will also address the fact that the differences in mechanical behaviors of Ailsa Craig and Trefor suites remain important future topics regarding the damage evolution of curling stones: "[...] With respect to curling stones, four different varieties exist (Ailsa Craig Blue Hone, Ailsa Craig Common Green, Blue Trefor, and Red Trefor), but their mechanical behaviors were not differentiated in this contribution. As such, further work is warranted to differentiate the dynamic behaviors of different curling stone types in order to understand how mineralogy and grain-size distribution influence the mechanical properties of granitoids, as well as how this leads to the accumulation of damage during repeated dynamic loading and Hertzian loading (e.g., Leung, 2020; Leung and McDonald, 2022; Padture and Lawn, 1995)."

*It would also be helpful to clarify how pristine rock samples are identified, which ones have experienced greater loading, and what information is available regarding samples/rocks that were previously used extensively (or used in varying numbers of games) prior to testing. I believe, including this context would strengthen the narrative and better support the findings.*

- Pristine rock samples of Ailsa Craig Blue Hone were derived from natural samples (that had not been made into curling stones). In terms of the rocks used

in the on-ice experiment, it is mentioned on l. 84–86 that the stones were used for less than one season. Unfortunately, in the case of the aged curling stones that were analyzed in the macroscopic and microscopic damage characterization sections, there is no information on the number of games that those rocks have been played in. We can only estimate the number of collisions over a lifespan of 10–15 years, which would be 29,000 on a conservative basis, or around 320 impacts per centimetre (please refer to Leung, 2025, https://doi.org/10.5194/egusphere-2025-3499-AC1, for calculations and assumptions).

*That said, I find this to be a very compelling and valuable piece of work!*
- Thank you – we greatly appreciate your constructive and encouraging feedback.

*Some more precise comments follow:*

1. *Is it rock physics or mostly rock mechanics for the title?*
   - We agree that rock mechanics (i.e., the study of the mechanical behavior of rocks under various stress conditions) is a more suitable phrase than rock physics (i.e., the study of the relationship between physical properties of rocks and their geophysical responses). This will be changed in the title and text.

2. Is there any difference between Ailsa Craig Common Green and Blue Home in terms of the structure, minerals, grain size etc.?
   - Ailsa Craig Common Green and Blue Hone are interpreted to represent the core and chill-margin facies (respectively) of the Ailsa Craig intrusion (Harrison et al., 1987). As such, the two rock types have essentially the same chemistry and mineralogy. Texturally, the two are quite different, with Ailsa Craig Common Green containing mm-sized spherical druses of quartz, whereas Ailsa Craig Blue Hone is more equigranular (with the exception of sparse alkali feldspar microphenocrysts, present in both). These aspects are detailed in Leung and McDonald (2022). The microfracture characterization focuses solely on Ailsa Craig Blue Hone, and comparisons of the mechanical properties of the different rock types is out of the scope of this contribution.

3. *Lines 118-119: Could you visualise by eyes the rock powder? What was the advantage using the high-speed camera instead, if the powder was retrieved post experiment? Which type of curling stone was used during these experiments (Common Green or Blue Hone)? Have you observed any differences in the quantity of powder generated and the microstructure (if you have tested both Common Green and Blue Hone)?*
   - The primary intention of the high-speed camera was to determine the contact time of the curling stone impacts. The visualization of the rock powder was a secondary (unintended) observation that led to the analysis of the rock powder by SEM methods. In the absence of the high-speed

camera, the rock powder would have likely been missed, as it was not visible in the GoPro footage.
- Only Ailsa Craig Common Green was used for the on-ice experiments, mainly due to the availability of curling stones at Curl Edinburgh. Ailsa Craig Blue Hone is no longer used in striking bands of contemporary stones (see Leung and McDonald 2022), so it is more difficult to test this question. Although the quantity of powder is a very intriguing question, it would be difficult to address, given the variable damage states of the stones and the exact collisional parameters (i.e., collisional velocity).

4. *Section 3.3: What about the Ailsa Crag Common Green stone? Why not used for thin sections? Is it Crag or Craig (see line 75)?*
    - Although thin sections were also prepared for Ailsa Craig Common Green samples, we ultimately chose to focus on the Ailsa Craig Blue Hone stones due to the greater availability for damaged samples. Thank you also for flagging the spelling errors. The correct spelling is Craig, and all incorrect instances of "Crag" have been corrected.

5. What about the rest 21 experiments? Is there any reason why were not analysed/presented herein?
    - Please refer to our response to Reviewer 1's comments regarding l. 171 for a description of our reply and proposed changes (Leung, 2025, https://doi.org/10.5194/egusphere-2025-3499-AC1, p. 10).

6. *On-ice experiments: what about lower impact velocities (~0.5 ms/-1)? Do you have an example that can provide a narrative/argument like that of hight velocities?*
    - Curling stones will deviate laterally from their intended trajectories ("curling"). At low velocities (e.g., ~ 0.5 $ms^{-1}$), this deviation is more substantial and experimentally challenging to control. Unfortunately, none of the high-speed camera experiments at low impact velocities were usable (see Leung, 2025, https://doi.org/10.5194/egusphere-2025-3499-AC1, p. 10). Although these low-velocity impacts could be achieved by pushing the stones into each other (rather than delivering them from 38 m away, as a curler would typically do), the primary intent of the study was to study damage accumulation in curling stones as an end in itself. As such, we decided to deliver the stones as true to the sport as possible, so this was ultimately not conducted.

7. *How have you defined the impact velocity range (0.5-2.9 ms-1)? I mean why this range?*
    - It is described on ll. 91–94 of the preprint that the common velocities for curling stone impacts are typically measured by recording the time that the stones cross two lines (called the hog lines). From my (DDVL) experience as a competitive curling athlete, typical hog-to-hog times for takeout shots range between 12–6.5 s, corresponding to measured velocities between

0.5–2.9 ms$^{-1}$. In actuality, the velocity can be as low as 0 (as defined by two curling stones just in contact, as in Exp. 0.4) and higher than 2.9 ms$^{-1}$, although velocities higher than 2.9 ms$^{-1}$ are not common.

8. *5: My understanding is that all these are different experiments. Which of the two materials were used herein (Common Green and Blue Hone)? Is there any info about the history of deformation, i.e., which cycle each of those images show?*
   ● Fig. 5 corresponds to the on-ice experiments, all of which were conducted with Ailsa Craig Common Green striking bands. This detail will be added to the caption of Fig. 5 to improve the clarity of the figure. Each image shows a different impact velocity. As it is difficult to replicate an impact on exactly the same impact area, cyclic testing was beyond the scope of this study. Additionally, we will add a note describing which rock types are used for which analyses of the study (see comment on p. 1 regarding proposed modifications to ll. 75–79 of the preprint).

9. *6: Does the Common Green stone show similar type of fractures?*
   ● Yes, Ailsa Craig Common Green can show similar types of fractures (see Fig. 10 in Leung and McDonald 2022), although anecdotally the fractures seem to penetrate less deep as compared to Ailsa Craig Blue Hone. It is uncertain (and unfortunately, beyond the scope of the present study) whether this anecdotal observation reflects a difference in mechanical behaviors between the two Ailsa Craig varieties or just represents an apparent effect due to the ages of the stones.

10. *Is Fig. 6 and 7 from the same sample?*
   ● No, Fig. 6 and 7 are not from the same sample, although they display similar crescent-shaped fractures. This is noted by the different sample numbers in the respective figure captions.

11. 8: How the grey and colour lines correlate – i.e. which data set is the coloured one if the whole dataset is in grey colours? What does 'n' stand for? 8e: what do the small circles represent? Which is the damage state – graph showing theta?
   ● It is noted in the figure caption that the grey dataset represents the whole dataset. "n" stands for the number of observed fractures, and the small circles represent outliers; these will be included in the figure caption for Fig. 8 (thank you for suggesting this addition). To improve the readability of the graphs in subfigure (e), we will modify the figure caption to specify the graph for r (left graph) and theta (right graph).

12. *Line 218: pls erase second have.*
   ● Thank you for flagging this typographical error – it has been corrected.

13. *9: What does the length (i.e., 30 cm) stand for? How do you define maturity? What does 'n' stand for?*
   ● The length (30 cm) is a scale bar.

- Maturity is defined on ll. 219–222 based on the observed development and depth of exhumation of the crescent-shaped fractures.
- "n" stands for the number of total observed fractures and will be added to the figure caption.

14. *10: How do you know that some of these fractures were not pre-existing (existing before the experiment), or else pre-existing fractures have not further propagated? Which of the two Ailsa Craig rocks is the one in Fig. 10?*
- Curling stones are manufactured such that large-scale fractures are rejected. Some of the smaller-scale fractures could certainly be pre-existing fractures. Figs. 11 and 12 better explore the possibility of pre-existing fractures.
- To improve the clarity of the text, we will add to the caption for Fig. 10 that this sample corresponds to Ailsa Craig Blue Hone.

15. *Lines 271: how do you identify pristine and juvenile stages?*
- Maturity is defined on ll. 219–222 based on the observed development and depth of exhumation of the crescent-shaped fractures. Pristine is not explicitly defined in this section, and as such we will include a definition of pristine following its first mention on l. 150. The pristine samples used here are natural samples that were not manufactured into curling stones. The incipient damage state refers to "partially formed fractures, which may be formed in segments", whereas juvenile refers to "fully linked crescent-shaped fractures, with 3D fracture surface hidden".

16. *Line 381: why rock physics and not rock mechanics?*
- This has been modified according to our response to item 1 of this review.

17. *Line 385: The damage evolution of curling stones consists of several damage states: pristine, weakly incipient, incipient, juvenile, and mature. How are those stages defined? If the same curling stone was used for previous games, how do you define the pristine sample? – I think I'm missing something here.*
- Maturity is defined on ll. 219–222 based on the observed development and depth of exhumation of the crescent-shaped fractures (see also reply to comment 15).
- With respect to the conceptual damage evolution model, the "pristine" damage state refers to a more broad state where no macrofractures are observed, representing somewhere between a new stone and a stone that has begun to develop weakly incipient crescent-shaped fractures. There is, of course, expected to be some degree of damage (i.e., flattening of the striking band and ejection of rock powder, as well as pitting), but there is otherwise no visible macroscopic evidence of fracturing.

18. *Lines 385-410: it would be great to support this narrative with images you have presented before in the results (or the accompanied material).*

- Although Fig. 15 is conceptual, the illustrations are based on actual profiles of rocks analyzed in this study. We feel that the addition of extra images would confuse readers rather than enhancing the figure. As such, we have decided not to adopt this suggestion, although we appreciate the rationale behind it.
- To facilitate the transition to the conceptual model for damage evolution, we will incorporate our reply to Reviewer 1 (Leung, 2025, https://doi.org/10.5194/egusphere-2025-3499-AC1, pp. 2–4) regarding cyclic Hertzian damage experiments on silicon carbide by Padture and Lawn (1995) which complement our results.

19. *I can see a damage evolution – can you please elaborate more on the actual model? I would possibly name it conceptual model.*
    - We have renamed the subsection to read "Conceptual model for damage evolution in curling stones".

20. *Conclusions- they need a bit of extra work and some narrative instead only bullet points.*
    - We will reformat the conclusions to read as a narrative. Additionally, we will include the limitations of the manuscript to highlight potential areas for future work. Please refer to Leung (2025, https://doi.org/10.5194/egusphere-2025-3499-AC1, p. 1) and pp. 1–2 of this response for the proposed additions regarding the limitations of the manuscript.
    - The modified conclusions section will now read as: "In this contribution, we have shown that curling stones are momentarily stressed to at least 300–680 MPa for high-velocity impacts (2.93 ± 0.15 ms$^{-1}$), exceeding the threshold for fatigue damage during dynamic loading. The impacts are shown to be dynamic in nature, as evidenced by (1) high strain rates (24 ± 4 s$^{-1}$) that approach those of co-seismic rock pulverization; (2) the ejection of rock powder as observed by the on-ice experiments; and (3) the presence of striations on crescent-shaped fractures that are interpreted to have formed by dynamic microfracture propagation (mirror-mist-hackle pattern). Crescent-shaped fractures are interpreted to be Hertzian cone fractures on the basis of their relationship to contacts between colliding curling stones, as well as their conoid morphology that is reminiscent of Hertzian cone fractures observed in other natural and engineered materials. We interpret crescent-shaped fractures to develop early via high-velocity impacts, in which damage initiates mainly as intragranular microcracks in feldspars, which propagate into transgranular microcracks that eventually develop into through-going microfaults as crescent-shaped fractures. Subsequent impacts propagate and coarsen these crescent-shaped fractures, and a localized damage zone develops in the collet between the crescent-shaped fractures and the striking band."

*All the best,*

*Elma Charalampidou*

Thank you again for your time and for your constructive feedback, which will serve to strengthen the manuscript.

References cited in this reply:

- Harrison, R. K., Stone, P., Cameron, I. B., Elliot, R. W., and Harding, R. R.: Geology, petrology and geochemistry of Ailsa Craig, Ayrshire, 1987.
- Leung, D. D. V.: "Reply to RC1", https://doi.org/10.5194/egusphere-2025-3499-AC1, 2025.
- Leung, D. D. V.: Where curling collides with rock physics: Characterising the damage evolution of curling stones, MScR thesis, University of Edinburgh, United Kingdom, 2020.
- Leung, D. D. V. and McDonald, A. M.: Taking rocks for granite: An integrated geological, mineralogical, and textural study of curling stones used in international competition, The Canadian Mineralogist, 60, 171–199, https://doi.org/10.3749/canmin.2100052, 2022.
- Padture, N. P. and Lawn, B. R.: Contact Fatigue of a Silicon Carbide with a Heterogeneous Grain Structure, Journal of the American Ceramic Society, 78, 1431–1438, https://doi.org/10.1111/j.1151-2916.1995.tb08834.x, 1995.